# Spiked-CFR: Causal Representation Learning from LLMs via Wasserstein Projection Pursuit

Fan Wang [1]   Hengyu Yue [2]   Bowen Yu [1]   Weiming Liu [1]   Zongxin Yang [3]   Xuyun Zhang [4]   Xiaolin Zheng [1]
Chaochao Chen [1]   Shuiguang Deng [1]

## Abstract

Estimating treatment effects from observational text is increasingly practical with Large Language Models (LLMs). However, applying causal representation learning directly to high-dimensional LLM embeddings faces a fundamental barrier: empirical Wasserstein matching suffers from the *curse of dimensionality*, rendering standard generalization guarantees effectively vacuous. We propose SPIKED-CFR, a framework bridging this gap by assuming a *Spiked Structure*, where treatment selection bias is assumed to manifest primarily as a low-dimensional treated–control discrepancy in the semantic representation. We develop Wasserstein Projection Pursuit, a minimax objective that adversarially learns an orthogonal projection on the Stiefel manifold to identify and balance only this subspace while preserving prognostic information. Under a *spiked structure*, we show the projected discrepancy can be estimated at a rate governed by the intrinsic dimension $k \ll D$, and we derive a tighter PEHE generalization bound that depends on $k$ rather than the ambient embedding dimension. Experiments on four semi-synthetic benchmarks and four real-world clinical benchmarks demonstrate improved accuracy and robustness over strong baselines.

## 1. Introduction

Estimating causal effects from observational data is crucial to decision-making in fields such as healthcare, education,



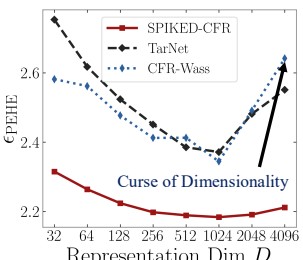 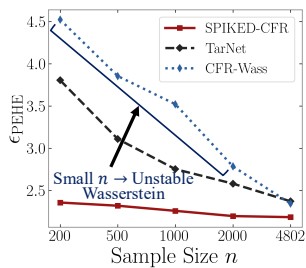

(a) $\epsilon_{\text{PEHE}}$ vs. dimension $D$     (b) $\epsilon_{\text{PEHE}}$ vs. sample size $n$

*Figure 1.* Empirical results on ACIC (measured by $\epsilon_{\text{PEHE}}$, lower is better) illustrate the fragility of Wasserstein-based balancing for LLM representations. (a) Varying the representation dimension $D$ shows that small $D$ induces an information bottleneck for LLM embeddings, while CFR-Wass degrades again at large $D$, consistent with the curse-of-dimensionality in Wasserstein-based balancing. (b) Varying sample size $n$ further shows that Wasserstein-based balancing becomes unstable under small-$n$. In contrast, our SPIKED-CFR remains stable across both $D$ and $n$.



and public policy (Liu et al., 2025a; Chen et al., 2024; Imai & Nakamura, 2024). In many real applications, however, the key confounders are not stored in structured tabular data. Instead, they are written in the free-form text, e.g., clinical notes, social media discussions (Roberts et al., 2020; Verma et al., 2025; Feder et al., 2023). This motivates an increasingly important goal: *end-to-end* causal effect estimation directly from unstructured text under the potential outcomes framework (Rubin, 2005).

Recent advances in large language models (LLMs) have made this goal feasible in practice (Veljanovski & Wood-Doughty, 2024; Ma et al., 2026; Huynh et al., 2025; Zhang et al., 2024). A representative example is NATURAL (Dhawan et al., 2024), which prompts an LLM to identify causal variables from text and to output propensity and outcome probabilities for downstream estimation. While promising, such pipelines can be fragile: they rely on strong assumptions about the stability and calibration of LLM probability outputs, which may vary with prompting and Reinforcement Learning from Human Feedback (RLHF)-style tuning (Kiciman et al., 2023; Zečević et al., 2023). This suggests an alternative direction: keep the semantic representation produced by an LLM, and perform causal inference directly within the resulting representation space.

[1]College of Computer Science and Technology, Zhejiang University, Hangzhou, China [2]College of Computer Science and Technology, China University of Petroleum (East China), Qingdao, China [3]Department of Biomedical Informatics, Harvard Medical School, Boston, MA, USA [4]Department of Computing, Macquarie University, Sydney, NSW 2109, Australia. Correspondence to: Xiaolin Zheng <xlzheng@zju.edu.cn >.

*Proceedings of the 43rd International Conference on Machine Learning*, Seoul, South Korea. PMLR 306, 2026. Copyright 2026 by the author(s).

Within causal effect estimation, this direction naturally connects to causal representation learning. Among many approaches, Counterfactual Regression (CFR) (Shalit et al., 2017) is a foundational framework. To estimate the Conditional Average Treatment Effect (CATE), CFR learns a representation that is predictive of outcomes while balancing the covariate distributions between treated and control groups using an Integral Probability Metric (IPM, often instantiated as the Wasserstein distance) (Hassanpour & Greiner, 2019). In the context of modern LLMs, a natural strategy is to treat high-dimensional text embeddings as covariates and apply CFR end-to-end.

However, directly balancing high-dimensional LLM embeddings faces a principled statistical barrier: the *curse of dimensionality* in distribution matching. In particular, the empirical Wasserstein distance between two distributions in $\mathbb{R}^D$ can converge as slowly as $n^{-1/D}$ (Fournier & Guillin, 2015; Niles-Weed & Rigollet, 2022). For typical LLM embeddings where $D$ is on the order of thousands (e.g., ~4096), this rate is effectively zero at realistic sample sizes. As a result, Wasserstein-based balancing in the full embedding space may overfit finite-sample noise instead of capturing true distribution shifts (as illustrated in Figure 1).

One might hope to address this by employing established high-dimensional OT variants, such as Sliced Wasserstein (Kolouri et al., 2019) or entropic regularization (Cuturi, 2013). While valuable, these tools are insufficient for two reasons. First, methods like Sliced Wasserstein average discrepancies over random projections, which can dilute sparse confounding signals. Consequently, they may miss the worst direction where treated and control groups differ most, which is likely where confounding can hide. Second, generic global balancing can lead to over-balancing: forcing the distributions to match in all directions may inadvertently discard prognostic information (features predictive of the outcome but unrelated to treatment assignment), thereby increasing the variance of CATE estimates (Assaad et al., 2021; Zhang et al., 2020). For instance, a text embedding may entangle a few confounding dimensions that govern treatment assignment with other dimensions that are purely outcome-predictive; aligning the entire space can unnecessarily distort the latter. Ideally, we seek a method that identifies and balances the low-dimensional subspace most likely to carry confounding-related imbalance, while preserving the rich prognostic semantics captured by the LLM.

To address this challenge, we make a simple but useful structural assumption: although LLM text embeddings are high-dimensional, the treated–control imbalance most relevant to confounding adjustment concentrates in a low-dimensional subspace (the "spike"). This insight is motivated by the causal intuition that treatment selection often depends on a small subset of latent factors, while the remaining varia-

tion captures shared semantics that can still be predictive of outcomes. We call this the *spiked structure*. Building on this insight, we propose SPIKED-CFR, an end-to-end framework that performs worst-case low-dimensional balancing over frozen LLM embeddings. We develop Wasserstein Projection Pursuit (WPP) as a min–max objective: an adversary searches for an orthogonal projection onto a subspace of dimension $k \ll D$ that maximizes the distributional discrepancy, thereby identifying the most imbalanced subspace that is most likely to capture confounding-relevant imbalance. Meanwhile, the representation learner minimizes this worst-case discrepancy jointly with outcome prediction. Under the spiked assumption, we show that this projected discrepancy recovers a convergence rate of $O(n^{-1/k})$, bypassing the curse of dimensionality.

Our contributions are as follows:

1. **Framework:** We propose SPIKED-CFR, which combines frozen LLM text embeddings with a worst-case projected Wasserstein balancing objective with CFR-based causal representation learning.

2. **Theory:** Under spiked structure, we show that the projected discrepancy converges with $k$ rather than $D$ ($k \ll D$), and derive a tighter PEHE upper bound for high-dimensional representations.

3. **Benchmarks:** Motivated by the lack of text-based counterfactual benchmarks, we construct prompt-generated variants of widely used semi-synthetic datasets, i.e., IHDP (Hill, 2011) and ACIC (Dorie et al., 2019), enabling systematic evaluation with ground-truth effects from natural-language inputs. We release the processed IHDP-post and ACIC-post benchmarks at https://github.com/Aurora-Brooks/Spiked-CFR-Benchmarks.

4. **Experiments:** We evaluate SPIKED-CFR on eight post-based benchmarks spanning semi-synthetic datasets and real-world clinical treatment comparisons; SPIKED-CFR achieves the strongest overall performance among text-based baselines, and our analyses support stable balancing in high-dimensional LLM representations.

**Conflict of Interest Disclosure** The authors declare no financial conflicts of interest related to this work.

## 2. Related Work

**Causal Inference with Large Language Models.** Work combining NLP and causal inference has evolved from treating text as noisy proxies (Pryzant et al., 2021; Roberts et al., 2020) to learning representations for confounding adjustment (Veitch et al., 2020; Zhou & He, 2023). Recent progress in LLMs has led to two complementary paradigms (Liu et al., 2025a; Ma, 2025; Chen et al., 2023; Wang et al., 2025b): *LLM-as-Estimator* uses an LLM to directly pre-

dict propensity scores or counterfactual outcomes (Dhawan et al., 2024; Veljanovski & Wood-Doughty, 2024), whereas *LLM-as-Extractor* uses an LLM to extract/encode information from text and performs effect estimation with a downstream causal learner (Pryzant et al., 2021; Veitch et al., 2020). While estimator-style methods can be effective, they typically rely on well-calibrated LLM outputs and can be sensitive to prompting; recent studies report systematic errors and overconfidence on causal reasoning tasks (Joshi et al., 2024; Kiciman et al., 2023). We therefore follow the extractor paradigm: we treat LLM as a semantic encoder and perform causal adjustment with a principled statistical objective.

**Balancing high-dimensional representations.** Standard balancing objectives in causal representation learning, e.g., Wasserstein matching (Shalit et al., 2017; Yan et al., 2024), suffer from the curse of dimensionality on LLM embeddings ($D \approx 4096$): empirical OT can converge as slowly as $n^{-1/D}$ (Fournier & Guillin, 2015), making full-space balancing ineffective at realistic sample sizes. With spiked transport model (Niles-Weed & Rigollet, 2022), we use Wasserstein projection pursuit (Paty & Cuturi, 2019; Lin et al., 2020) to focus discrepancy estimation on an adversarially chosen low-dimensional subspace (Stiefel-constrained), rather than averaging over random projections (Kolouri et al., 2019).

## 3. Preliminaries

**Problem Setup.** We consider the problem of estimating Conditional Average Treatment Effect (CATE) from observational data within the potential outcomes framework (Rubin, 2005). Let $T \in \{0, 1\}$ denote the binary treatment assignment, and $Y \in \mathbb{R}$ denote the observed outcome. Corresponding to the treatment assignment, let $Y(1)$ and $Y(0)$ be the potential outcomes under the treated and control scenarios, respectively. The observed outcome is given by $Y = TY(1) + (1 - T)Y(0)$. We assume access to a set of covariates $X \in \mathcal{X}$, which confound the relationship between $T$ and $Y$. Our objective is to estimate CATE, defined as $\tau(x) = \mathbb{E}[Y(1) - Y(0) \mid X = x]$. Identification relies on the standard assumptions of *consistency* (i.e., $Y = Y(T)$) and *strong ignorability* (i.e., *unconfoundedness* $(Y(0), Y(1)) \perp\!\!\!\perp T \mid X$ and *overlap* $0 < \mathbb{P}(T = 1 \mid X = x) < 1, \forall x \in \text{supp}(X)$).

**Representation Learning for CATE Estimation.** Standard supervised learning typically estimates conditional outcome $\mathbb{E}[Y|X, T]$ directly. However, the distribution of covariates often differs between treatment groups, i.e., $P(X \mid T = 1) \neq P(X \mid T = 0)$, a phenomenon known as selection bias (or covariate shift) (Rosenbaum & Rubin, 1983; Shimodaira, 2000). Consequently, direct regression on $X$ often yields unreliable counterfactual estimates due

to extrapolation errors in regions of limited overlap. To address this, we follow the Counterfactual Regression (CFR) framework (Shalit et al., 2017). The core idea is to learn a representation map $\Phi : \mathcal{X} \to \mathbb{R}^D$ and a hypothesis $h$ such that the induced distributions $P_\Phi^t = P(\Phi(X) \mid T = t)$ are balanced, while preserving sufficient information for predicting outcomes. Formally, this learning objective is motivated by minimizing an upper bound on Precision in Estimating Heterogeneous Effects (PEHE) error (Hill, 2011), which measures $L_2$ error of the estimated CATE. As Shalit et al. (2017) shows, the bound takes the form:

$$\epsilon_{\text{PEHE}} \leq \epsilon_F(h, \Phi) + \alpha \cdot \text{IPM}(P_\Phi^1, P_\Phi^0) + \text{Const}, \quad (1)$$

where $\epsilon_F$ is the factual prediction loss, $\text{IPM}(\cdot, \cdot)$ is an Integral Probability Metric (Sriperumbudur et al., 2012; Müller, 1997) measuring distribution discrepancy, and $\alpha$ is a weighting hyperparameter trading off prediction and balance (see Appendix A for more formal definitions). Wasserstein distance is a theoretically robust and widely adopted choice for the IPM in Eq. (1) as it captures the geometry of embedding space (Shalit et al., 2017; Assaad et al., 2021). Therefore, in this work, we instantiate IPM as the $p$-Wasserstein distance, i.e., $\text{IPM}(P_\Phi^1, P_\Phi^0) = W_p(P_\Phi^1, P_\Phi^0)$. Although Wasserstein-based balancing has been successfully applied in causal representation learning (Li et al., 2022; Wang et al., 2024), its direct application in high-dimensional text representations faces statistical challenges, which we address in Section 4.

## 4. Methodology

We propose SPIKED-CFR, an end-to-end framework for estimating CATE from unstructured text. We first describe the overall pipeline (Section 4.1), then motivate why Wasserstein balancing becomes statistically vacuous in high-dimensional LLM spaces (Section 4.2). We next introduce a projection-pursuit Wasserstein discrepancy under Spiked Structure (Section 4.3), and finally present the resulting min–max objective and Stiefel-manifold optimization (Section 4.4).

### 4.1. Framework Overview

Figure 2 summarizes SPIKED-CFR as an end-to-end pipeline with four modules:

i. **LLM-based Extraction.** Let $\mathcal{D} = \{W_i\}_{i=1}^n$ be a dataset of raw posts. We parse each $W_i$ into $(T_i, Y_i, \tilde{X}_i)$ via prompting, where $\tilde{X}_i$ is the pre-treatment text defined to contain confounding information while excluding outcome-revealing content to reduce leakage. Prompt templates and post-processing rules are provided in Appendix D.5.

ii. **High-Dimensional Representation Learning.** We en-

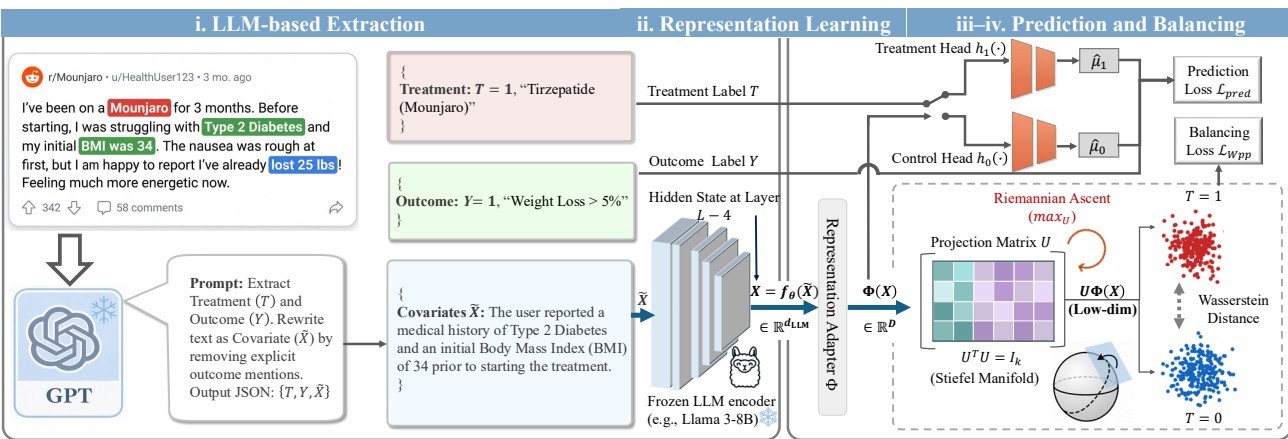

*Figure 2.* Overview of the SPIKED-CFR framework. The pipeline comprises four stages: (1) **LLM-based Extraction**: extracting structured variables $(T, Y, \tilde{X})$ from raw text $W$; (2) **Representation Learning**: encoding text via a frozen LLM ($f_\theta$) and a trainable adapter ($\Phi$); (3) **Outcome Prediction**: estimating potential outcomes via split-heads ($h_0, h_1$); and (4) **Adversarial Balancing**: mitigating selection bias via a min-max game, where the adversary $U$ **maximizes** the Wasserstein discrepancy while the representation learner minimizes it. Note that $U$ only affects the balancing regularization; outcome prediction utilizes the full representation $\Phi(X)$.

code the extracted pre-treatment text $\tilde{X}_i$ with a frozen LLM $f_\theta$ and obtain $X_i = f_\theta(\tilde{X}_i)$, using the hidden state from the 4th-to-last layer (L-4). We assume the extracted pre-treatment text $\tilde{X}_i$ (and thus $X_i = f_\theta(\tilde{X}_i)$) is sufficient for adjustment, i.e., $(Y_i(0), Y_i(1)) \perp\!\!\!\perp T_i \mid X_i$. We then learn a trainable adapter $\Phi : \mathbb{R}^{d_{\text{LLM}}} \to \mathbb{R}^D$ and use $\Phi(X_i)$ for outcome prediction. We keep $D$ in the order of thousands to preserve semantic expressiveness (see Figure 1a). However, this choice makes estimating Wasserstein discrepancies in $\mathbb{R}^D$ statistically challenging; we will analyze this issue in Sec. 4.2.

iii. **Outcome Prediction.** We use treatment-specific heads $h_0, h_1$ to model conditional mean potential outcomes $\hat{\mu}_t(X_i) = h_t(\Phi(X_i))$ for $t \in \{0, 1\}$. The estimated CATE is $\hat{\tau}(X_i) = \hat{\mu}_1(X_i) - \hat{\mu}_0(X_i)$.

iv. **Adversarial Balancing via Wasserstein Projection Pursuit (WPP).** To mitigate selection bias, we balance the treated/control representation distributions induced by $\Phi(X_i) \in \mathbb{R}^D$. Directly matching these distributions in $\mathbb{R}^D$ is statistically inefficient when $D$ is large. Instead, we introduce an orthogonal projection matrix $U \in \mathbb{R}^{k \times D}$ with $UU^\top = I_k$ ($k \ll D$) and measure discrepancy using Wasserstein distance between projected representations $U\Phi(X_i)$. Training follows a min–max game: predictor $(\Phi, h_0, h_1)$ minimizes the factual loss plus a discrepancy penalty, while the adversary $U$ maximizes the projected Wasserstein discrepancy (optimization details in Section 4.4).

### 4.2. Statistical Limitations of Standard CFR

A critical challenge in applying CFR to LLM embeddings is that the discrepancy term in PEHE bound (Eq. (1)) is *population* quantity, e.g., $W_p(P_\Phi^1, P_\Phi^0)$. In practice, we only observe finite samples and must approximate this term using *empirical* Wasserstein distance. In high-dimensional em-

bedding spaces, this approximation suffers from the curse of dimensionality. To formalize this, we first recall a standard lower bound for estimating Wasserstein distances in $\mathbb{R}^D$:

**Proposition 4.1** (Curse of Dimensionality (Fournier & Guillin, 2015; Niles-Weed & Rigollet, 2022))**.** *Let $\hat{P}_n$ and $\hat{Q}_n$ denote the empirical measures formed from $n$ i.i.d. samples of $P$ and $Q$ in $\mathbb{R}^D$ ($D > 2p$). Then the expected estimation error of the empirical Wasserstein distance is lower-bounded by*

$$\mathbb{E}\left[\left|W_p(\hat{P}_n, \hat{Q}_n) - W_p(P, Q)\right|\right] \gtrsim n^{-1/D}. \quad (2)$$

To connect this to CFR, consider the decomposition of the population discrepancy via the triangle inequality:

$$W_p(P_\Phi^1, P_\Phi^0) \leq \widehat{W}_p + \Delta_{n,D},$$
$$\Delta_{n,D} := \left|W_p(\hat{P}_\Phi^1, \hat{P}_\Phi^0) - W_p(P_\Phi^1, P_\Phi^0)\right|, \quad (3)$$

where $n := \min\{n_0, n_1\}$ is the number of samples in the smaller group. Substituting decomposition into the PEHE bound (Eq. (1)) yields the following corollary, which highlights the failure of standard balancing in high dimensions.

**Corollary 4.2** (Vacuity of Wasserstein-based CFR in High Dimensions)**.** *Instantiating the IPM term in Eq. (1) with $W_p$, Then for any hypothesis $h$ and representation map $\Phi$, the PEHE error is bounded by:*

$$\epsilon_{\text{PEHE}}(h, \Phi) \leq \epsilon_F(h, \Phi) + \alpha \widehat{W}_p + \alpha \Delta_{n,D} + \text{Const.} \quad (4)$$

*Moreover, in the worst case, the estimation error satisfies:*

$$\mathbb{E}\left[\Delta_{n,D}\right] \gtrsim n^{-1/D}. \quad (5)$$

*A derivation of Eq. (4) is provided in Appendix B. Consequently, even if an algorithm successfully minimizes the*

*empirical discrepancy to zero ($\widehat{W}_p \approx 0$), the generalization bound in Eq. (4) remains dominated by the non-vanishing term of order $\Omega(n^{-1/D})$, rendering the theoretical guarantee vacuous for large $D$.*

**Remark.** The rate $n^{-1/D} = \exp(-(\log n)/D)$ decreases very slowly in high dimensions. Thus, Wasserstein-based balancing can yield a nearly non-informative PEHE guarantee for LLM embeddings. Therefore, we seek a low-dimensional subspace in which the treated–control discrepancy can be reliably estimated, while preserving the predictive information in the full representation for outcome modeling. This dual requirement motivates our projection-pursuit balancing strategy in the next subsection.

## 4.3. Theoretical Foundations of SPIKED-CFR

To address the dimensionality challenge, we hypothesize that although $\Phi(X) \in \mathbb{R}^D$ is high-dimensional, the treated–control distribution shift most relevant to confounding adjustment concentrates on a low-dimensional set of latent factors. This is motivated by the causal intuition that selection bias often depends on a small subset of confounding-related factors. Accordingly, in the representation space, most dimensions capture variation that is shared across treatment groups, while remaining informative for modeling potential outcomes. We formalize this intuition via a *Spiked Structure*.

**Assumption 4.3** (Spiked Structure). Let $P_\Phi^t := \mathrm{Law}(\Phi(X) \mid T = t)$ denote the representation distribution in group $t \in \{0, 1\}$. Assume there is a $k$-dimensional subspace $\mathcal{S} \subset \mathbb{R}^D$ (the "spike") with $k \ll D$ such that

$$P_\Phi^t = \mathrm{Law}\Big(S^{(t)} + \varepsilon\Big), \quad t \in \{0, 1\}, \qquad (6)$$

where $S^{(t)} \in \mathcal{S}$ captures the group-dependent component (allowed to vary with $t$), and $\varepsilon \in \mathcal{S}^\perp$ captures shared variation with $\varepsilon \perp T$ (i.e., $\mathrm{Law}(\varepsilon \mid T = 1) = \mathrm{Law}(\varepsilon \mid T = 0)$). Here, $\mathrm{Law}(\cdot)$ denotes the distribution of a random variable.

**Wasserstein projection pursuit.** Under Assumption 4.3, the treated–control discrepancy concentrates on an unknown low-dimensional subspace. Rather than estimating $W_p(P_\Phi^1, P_\Phi^0)$ in $\mathbb{R}^D$, we measure discrepancy after projecting onto $k$ dimensions and use projection pursuit to search for the most discriminative subspace. Specifically, we define the projection-robust Wasserstein distance:

$$\mathcal{W}_{p,k}(P, Q) := \max_{U \in \mathcal{V}_k} W_p\big(U^\sharp P, \ U^\sharp Q\big), \qquad (7)$$

where $\mathcal{V}_k := \{U \in \mathbb{R}^{k \times D} : UU^\top = I_k\}$ is the Stiefel manifold (Absil et al., 2008), $U^\sharp P$ denotes the push-forward measure of $P$ under the linear map $z \mapsto Uz$. We enforce the orthogonality constraint ($UU^\top = I_k$) to ensure the projection preserves the geometric structure of the subspace

without arbitrary scaling distortion. Intuitively, the maximization searches for the $k$-dimensional subspace that amplifies the treated–control imbalance, thereby automatically identifying the confounding-relevant imbalance associated with selection bias. Crucially, this projected discrepancy can be estimated at a rate governed by $k$ rather than the ambient dimension $D$, as formalized in Proposition 4.4.

**Proposition 4.4** (Estimation Rate of WPP under the Spiked Structure (Niles-Weed & Rigollet, 2022)). *Assume Assumption 4.3 and standard regularity conditions detailed Appendix B. Let $\hat{P}_\Phi^t$ be empirical measures formed from $n_t$ i.i.d. samples from $P_\Phi^t$, set $n := \min\{n_0, n_1\}$. Then empirical WPP estimator $\widehat{\mathcal{W}}_{p,k} := \mathcal{W}_{p,k}(\hat{P}_\Phi^1, \hat{P}_\Phi^0)$ satisfies*

$$\mathbb{E}\big|\widehat{\mathcal{W}}_{p,k} - \mathcal{W}_{p,k}(P_\Phi^1, P_\Phi^0)\big| \lesssim n^{-1/k} + \sqrt{\frac{D \log n}{n}}. \ (8)$$

By integrating this rate into the CFR framework, we derive the main generalization bound for SPIKED-CFR:

**Theorem 4.5** (Generalization Bound of SPIKED-CFR). *Let $\hat{h}$ be the hypothesis and $\hat{\Phi}$ the representation learned by SPIKED-CFR. Under Assumption 4.3, the expected PEHE error is bounded by:*

$$\epsilon_{\mathrm{PEHE}}(\hat{h}, \hat{\Phi}) \leq$$

$$\epsilon_F(\hat{h}, \hat{\Phi}) + \alpha \widehat{\mathcal{W}}_{p,k} + \mathcal{O}\left(n^{-1/k} + \sqrt{\frac{D \log n}{n}}\right) + \mathrm{Const.}$$
$$(9)$$

*where $\alpha$ is the trade-off weight. A detailed derivation is provided in Appendix B.*

**Remark.** Unlike the standard CFR bound (Corollary 4.2), our bound on $\epsilon_{\mathrm{PEHE}}$ depends primarily on the *intrinsic dimension* $k$ (typically $k \ll D$), effectively mitigating the curse of dimensionality for the discrepancy estimate. The term involving $D$ decays as $\mathcal{O}(n^{-1/2})$, ensuring that SPIKED-CFR maintains a tight theoretical guarantee for treatment effect estimation even with high-dimensional LLM representations. This tighter bound directly matches our dual requirement: we estimate the treated–control discrepancy in a low-dimensional projection (making Wasserstein-based balancing statistically meaningful), while preserving the full representation $\Phi(X)$ for outcome modeling.

## 4.4. Objective Function and Optimization

**Training Objective.** Based on generalization bound in Theorem 4.5, we formulate the training of SPIKED-CFR as a min-max game. We aim to learn a representation $\Phi$ and hypothesis $h$ that minimize the factual prediction error while playing against an adversary $U$ that maximizes the distributional discrepancy. The total objective is:

$$\min_{\Phi, \{h_0, h_1\}} \max_{U \in \mathcal{V}_k} \mathcal{L}_{\mathrm{pred}}(\Phi, h) + \alpha \mathcal{L}_{\mathrm{WPP}}(\Phi, U). \ (10)$$

where $\alpha > 0$ trades off factual prediction and distribution balancing. The first term is the standard factual prediction loss over the observed outcomes:

$$\mathcal{L}_{\text{pred}}(\Phi, h) := \frac{1}{n} \sum_{i=1}^{n} \ell(h_{T_i}(\Phi(X_i)), Y_i). \quad (11)$$

where $\ell$ is mean squared error for continuous outcomes and cross-entropy for binary outcomes. The second term is our WPP discrepancy:

$$\mathcal{L}_{\text{WPP}}(\Phi, U) := W_p\left(U^{\sharp}\hat{P}_{\Phi}^1, U^{\sharp}\hat{P}_{\Phi}^0\right), \qquad U \in \mathcal{V}_k, \quad (12)$$

where $\mathcal{V}_k = \{U \in \mathbb{R}^{k \times D} : UU^{\top} = I_k\}$, $\hat{P}_{\Phi}^t$ is the empirical measure of representations in group $t$, and $U^{\sharp}$ is the push-forward under $z \mapsto Uz$.

**Alternating optimization.** Since Eq. (10) presents a non-convex optimization problem with manifold constraints, we adopt an alternating strategy: (i) *adversary ascent* on $U$ on the Stiefel manifold $\mathcal{V}_k$ using Riemannian gradient steps to increase $\mathcal{L}_{\text{WPP}}$, and (ii) *learner descent* on $(\Phi, h_0, h_1)$ to decrease $\mathcal{L}_{\text{pred}} + \alpha \mathcal{L}_{\text{WPP}}$. In practice, we compute $W_p$ using a differentiable Sinkhorn solver (Cuturi, 2013). The full training procedure, Stiefel-manifold optimization details and time complexity analysis are deferred to Appendix C.

## 5. Experiments

### 5.1. Experimental Setup

**Datasets.** We evaluate on eight post-based causal inference benchmarks: four semi-synthetic and four real-world. Following NATURAL (Dhawan et al., 2024), we use two semi-synthetic datasets (Hillstrom (Hillstrom, 2008) and RetailHero (Group, 2019)) where randomized tabular records are rendered into observational-style reports with an LLM, and four real-world Reddit datasets with ground-truth ATE provided by RCTs. We refer to the four clinical pairs as Sema vs. Tirz, Sema vs. Lira, Eren vs. Topi, and Onab vs. Topi. To enable CATE evaluation with counterfactual supervision, we additionally construct IHDP-post and ACIC-post by prompting an LLM to convert the standard IHDP (Hill, 2011) and ACIC (Dorie et al., 2019) tabular benchmarks into first-person posts. During generation, the LLM is conditioned on covariates, treatment, and the factual outcome only (the counterfactual outcome is hidden and used solely for evaluation) to avoid leakage of potential outcomes into the text. IHDP-post and ACIC-post contain 747 and 4,802 instances, respectively; only these two benchmarks provide individual-level counterfactual outcomes, while the remaining datasets are evaluated using ATE-based metrics against the ground truth reported in NATURAL (Dhawan et al., 2024). We provide full dataset statistics, the ground-truth sources for each benchmark, and the exact prompts

used to generate IHDP-post/ACIC-post in Appendix D. We release the processed IHDP-post and ACIC-post benchmarks at `https://github.com/Aurora-Brooks/Spiked-CFR-Benchmarks`.

**Baselines.** We compare against four families of estimators: *(i) Classical causal estimators on text features:* BOW-OLS; BERT-PSM (Rosenbaum & Rubin, 1983) on frozen BERT embeddings (Devlin et al., 2019); BERT-OForest (Oprescu et al., 2019)/BERT-CForest (Athey et al., 2019); and Sentence Encoder (SBERT) (Reimers & Gurevych, 2019). *(ii) Frozen-LLM representations with standard CATE networks:* Llama-CForest; Llama-TARNet (Shalit et al., 2017); Llama-Dragonnet (Shi et al., 2019); and Llama-CFR-MMD/Wass (CFR (Shalit et al., 2017) with MMD or Wasserstein IPM) on frozen Llama embeddings (Grattafiori et al., 2024). *(iii) Text-based causal effect modeling:* CausalBERT (Pryzant et al., 2021). *(iv) End-to-end LLM effect estimation:* NATURAL (Dhawan et al., 2024). We also report CFR-Wass (Structured) (Shalit et al., 2017) on the original tabular covariates as a non-text reference. All Llama-based baselines use the same frozen-embedding configuration and train/validation protocol as our method; details are in Appendix D.2.

**Metrics.** We evaluate both conditional and average treatment effect estimation (CATE and ATE). On datasets with counterfactual ground truth (IHDP-post and ACIC-post), we report $\epsilon_{\text{PEHE}}$ to measure CATE error, where lower is better. Across all datasets, we report the estimated ATE and its deviation from the ground truth: $\epsilon_{\text{ATE}}$ and RMSE(ATE), where lower values indicate more accurate ATE estimation, while ATE is better when closer to the ground truth. Formal definitions are provided in Appendix D.3.

**Implementation Details.** We use a 75/15/10 train/validation/test split, select the model that performs best on the validation set, evaluate it on the test set, and report mean±std over 5 runs. Posts are encoded with frozen Llama-3-8B embeddings using the L-4 hidden state as $X \in \mathbb{R}^{4096}$ (see Figure. 4 for the layer choice). We optimize Eq. (10) with Adam for $(\Phi, h_0, h_1)$ and Riemannian SGD for $U$, and compute $p=2$ Wasserstein via Sinkhorn (Cuturi, 2013). Dataset-specific hyperparameters and sweeps are in Appendix D.4, with sensitivity for $k$ and $\alpha$ in Appendix E.4.

### 5.2. Main Results

**Across-benchmark performance.** Table 1 and Table 2 summarize results on eight benchmarks: semi-synthetic datasets w/without counterfactual supervision and clinical pairs with RCT-grounded ATE. Overall, SPIKED-CFR delivers the strongest performance across both settings: it achieves the best estimation accuracy while producing ATE estimates close to the ground truth. Notably, the gains are

*Table 1.* Results on semi-synthetic datasets. We report ATE for all datasets, $\epsilon_{\text{PEHE}}$ on IHDP/ACIC, and RMSE(ATE) on Hillstrom/Retail-Hero. Best results are in bold and second-best results are underlined. Ground-truth effect values follow NATURAL (Dhawan et al., 2024). NATURAL assumes binary outcomes and is thus not evaluated on IHDP/ACIC.

| | Benchmarks w/ Counterfactual GT | | | | Benchmarks w/o Counterfactual GT | | | |
| --- | --- | --- | --- | --- | --- | --- | --- | --- |
| | IHDP | | ACIC | | Hillstrom | | Retail Hero | |
| | ATE | $\epsilon_{PEHE}\downarrow$ | ATE | $\epsilon_{PEHE}\downarrow$ | ATE (%) | RMSE$\downarrow$ | ATE (%) | RMSE$\downarrow$ |
| BOW-OLS | $3.68 \pm 0.24$ | $2.75 \pm 0.28$ | $2.74 \pm 0.16$ | $2.68 \pm 0.06$ | $3.78 \pm 1.06$ | 2.54 | $6.08 \pm 1.26$ | 3.03 |
| BERT-PSM | $3.40 \pm 0.19$ | $2.73 \pm 0.16$ | $3.00 \pm 0.13$ | $6.14 \pm 0.15$ | $5.14 \pm 0.87$ | 1.29 | $6.75 \pm 1.59$ | 3.78 |
| BERT-OForest | $3.49 \pm 0.09$ | $3.07 \pm 0.16$ | $3.12 \pm 0.02$ | $2.43 \pm 0.05$ | $3.44 \pm 0.66$ | 2.73 | $6.55 \pm 0.02$ | 3.23 |
| BERT-CForest | $3.64 \pm 0.04$ | $2.57 \pm 0.28$ | $3.10 \pm 0.03$ | $2.45 \pm 0.02$ | $4.70 \pm 0.09$ | 1.39 | $5.53 \pm 0.32$ | 2.23 |
| Sentence Encoder | $3.36 \pm 0.56$ | $2.88 \pm 0.26$ | $2.91 \pm 0.08$ | $2.47 \pm 0.09$ | $0.00 \pm 0.00$ | 6.09 | $1.97 \pm 1.62$ | 2.11 |
| Llama-CForest | $4.34 \pm 0.19$ | $\underline{2.50 \pm 0.21}$ | $2.94 \pm 0.03$ | $2.26 \pm 0.03$ | $5.75 \pm 0.09$ | $\underline{0.35}$ | $3.97 \pm 0.37$ | $\underline{0.75}$ |
| Llama-TARNet | $4.11 \pm 0.04$ | $2.65 \pm 0.18$ | $3.05 \pm 0.08$ | $2.37 \pm 0.08$ | $5.45 \pm 0.63$ | 0.90 | $3.75 \pm 1.38$ | 1.45 |
| Llama-Dragonnet | $4.00 \pm 0.18$ | $2.68 \pm 0.40$ | $2.93 \pm 0.09$ | $\underline{2.21 \pm 0.27}$ | $\underline{6.23 \pm 0.43}$ | 0.45 | $\underline{2.93 \pm 0.88}$ | 0.96 |
| Llama-CFR-MMD | $\underline{4.13 \pm 0.21}$ | $2.51 \pm 0.15$ | $\underline{3.13 \pm 0.05}$ | $2.27 \pm 0.03$ | $5.81 \pm 0.42$ | 0.50 | $4.49 \pm 1.52$ | 1.92 |
| Llama-CFR-Wass | $4.11 \pm 0.04$ | $2.65 \pm 0.18$ | $2.94 \pm 0.46$ | $2.32 \pm 0.44$ | $5.76 \pm 0.38$ | 0.50 | $2.72 \pm 2.06$ | 2.15 |
| CausalBERT | $3.34 \pm 0.69$ | $3.32 \pm 0.55$ | $2.68 \pm 0.06$ | $3.32 \pm 0.04$ | $3.00 \pm 5.53$ | 6.33 | $5.19 \pm 5.60$ | 5.90 |
| NATURAL | N/A | N/A | N/A | N/A | $6.23 \pm 0.39$ | 0.41 | $4.36 \pm 0.34$ | 1.09 |
| SPIKED-CFR (Ours) | $\mathbf{4.14 \pm 0.16}$ | $\mathbf{2.47 \pm 0.20}$ | $\mathbf{3.16 \pm 0.05}$ | $\mathbf{2.18 \pm 0.03}$ | $6.21 \pm 0.31$ | **0.33** | $3.19 \pm 0.71$ | **0.72** |
| CFR-Wass (Structured) | $4.38 \pm 0.01$ | $1.03 \pm 0.05$ | $3.43 \pm 0.09$ | $1.57 \pm 0.18$ | $6.01 \pm 0.12$ | 0.14 | $3.80 \pm 0.16$ | 0.51 |
| Ground Truth | 4.19 | – | 3.47 | – | 6.09 | – | 3.32 | – |

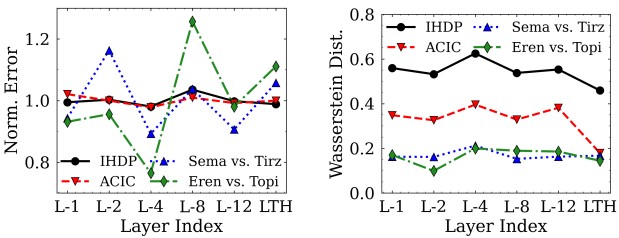

*(a)* SPIKED-CFR vs. RP    *(b)* SPIKED-CFR vs. PCA    *(c)* $\epsilon_{\text{PEHE}}$ on *ACIC*    *(d)* $\epsilon_{\text{ATE}}$ on *Eren vs. Topi*

*Figure 3.* **High-dimensional balancing.** Heatmaps show stable gains over RP/PCA across projection dimension $k$; raincloud plots show reduced seed variance (ACIC, Eren vs. Topi).

*(a)* Norm. Error vs. Layer.    *(b)* Wasserstein Dist. vs. Layer.

*Figure 4.* **Layer selection.** L-4 performs best and exposes a large projected Wasserstein gap.

more pronounced on the clinical tasks, where SPIKED-CFR attains the lowest RMSE(ATE) on all four drug-pair benchmarks. On the semi-synthetic benchmarks, SPIKED-CFR remains consistently competitive on $\epsilon_{\text{PEHE}}$ (IHDP/ACIC) and further improves RMSE(ATE) on Hillstrom and RetailHero. Across both tables, BERT-based methods are uniformly weaker than Llama-based ones, highlighting the advantage of modern LLM representations over BERT features for causal learning from text. CFR-Wass (Structured) on clean tabular covariates serves as an oracle reference, and

the remaining gap reflects the noise introduced by prompt-based text extraction in realistic end-to-end settings.

**What the baselines reveal about LLM causal estimation.**
While NATURAL is a strong recent approach, it is not uniformly optimal in our benchmarks and is restricted to binary outcomes, underscoring the practical fragility of relying on calibrated LLM probability outputs for IPW-style estimation. Meanwhile, among CFR variants on frozen LLM features, we observe that full-space Wasserstein matching can be unstable: Llama-CFR-Wass is sometimes worse than Llama-CFR-MMD, and can even degrade toward Llama-TARNet. These patterns are consistent with our motivation that Wasserstein-based balancing becomes statistically inefficient in very high dimensions, leaving residual imbalance under finite samples. SPIKED-CFR addresses this by focusing balancing capacity where confounding-relevant imbalance concentrates: it searches for the most shifted low-dimensional subspace (the "spike"), while outcome prediction uses the full $\Phi(X)$ to preserve prognostic semantics, yielding more reliable counterfactual estimates.

*Table 2.* Results on real-world clinical datasets. We report ATE and RMSE(ATE). Best results are in bold and second-best results are underlined. Ground-truth ATE values are taken from the matched RCTs as reported in NATURAL (Dhawan et al., 2024).

| | Sema vs. Tirz | | Sema vs. Lira | | Eren vs. Topi | | Onab vs. Topi | |
|---|---|---|---|---|---|---|---|---|
| | ATE (%) | RMSE ↓ | ATE (%) | RMSE ↓ | ATE (%) | RMSE ↓ | ATE (%) | RMSE ↓ |
| BOW-OLS | $3.14 \pm 3.37$ | 7.74 | $1.07 \pm 1.45$ | 15.84 | $14.59 \pm 1.27$ | 13.77 | $26.37 \pm 2.21$ | 14.80 |
| BERT-PSM | $-21.03 \pm 0.96$ | 31.15 | $12.46 \pm 0.36$ | 27.16 | $15.85 \pm 2.71$ | 12.74 | $19.28 \pm 0.28$ | 21.72 |
| BERT-OForest | $13.11 \pm 0.05$ | 3.00 | $1.62 \pm 0.29$ | 16.32 | $21.33 \pm 1.70$ | 7.17 | $22.73 \pm 0.09$ | 18.27 |
| BERT-CForest | $5.56 \pm 0.07$ | 4.55 | $-1.67 \pm 0.44$ | 13.04 | $19.30 \pm 0.15$ | 9.00 | $22.47 \pm 0.20$ | 18.53 |
| Sentence Encoder | $-8.31 \pm 4.41$ | 18.94 | $-3.83 \pm 1.71$ | 11.00 | $-0.81 \pm 0.52$ | 29.11 | $5.14 \pm 6.52$ | 36.45 |
| Llama-CForest | $8.24 \pm 0.32$ | 1.90 | $1.23 \pm 0.69$ | 15.94 | $18.72 \pm 0.38$ | 9.59 | $21.27 \pm 0.33$ | 19.73 |
| Llama-TARNet | $10.77 \pm 0.54$ | 0.85 | $-10.75 \pm 2.92$ | 4.91 | $24.83 \pm 1.63$ | 3.83 | $33.81 \pm 5.55$ | 9.08 |
| Llama-Dragonnet | $9.47 \pm 0.46$ | 0.79 | $-11.41 \pm 1.09$ | 3.47 | $15.16 \pm 0.62$ | 13.15 | $16.45 \pm 8.29$ | 25.91 |
| Llama-CFR-MMD | $9.02 \pm 1.74$ | 2.05 | $-4.33 \pm 1.80$ | 10.53 | $25.88 \pm 2.51$ | 3.49 | $32.63 \pm 5.09$ | 9.80 |
| Llama-CFR-Wass | $10.61 \pm 1.44$ | 1.52 | $-8.71 \pm 1.47$ | 6.17 | $26.65 \pm 3.82$ | 4.16 | $36.53 \pm 6.80$ | 8.14 |
| CausalBERT | $6.34 \pm 1.82$ | 4.19 | $-2.65 \pm 4.62$ | 12.91 | $10.81 \pm 2.00$ | 17.60 | $20.31 \pm 1.10$ | 20.72 |
| NATURAL | $8.78 \pm 0.70$ | 1.50 | $-16.33 \pm 1.54$ | 2.24 | $24.47 \pm 1.12$ | 3.99 | $42.87 \pm 1.87$ | 2.64 |
| SPIKED-CFR (Ours) | **$10.35 \pm 0.60$** | **0.65** | $-14.72 \pm 0.56$ | **0.56** | **$28.73 \pm 1.62$** | **1.68** | **$42.30 \pm 0.46$** | **1.38** |
| Ground Truth | 10.11 | | -14.70 | | 28.30 | | 41.00 | |

*(a)* Raw LLM    *(b)* CFR-Wass    *(c)* SPIKED-CFR (Ours)    *(d)* Treatment Classification AUC

*Figure 5.* **Overlap analysis.** SPIKED-CFR improves covariate balance: propensity mass shows stronger treated–control overlap and treatment predictability decreases (AUC ↓), relative to raw LLM and CFR-Wass.

## 5.3. Ablations and Analysis

**WPP: stability and robustness in high dimensions.** Figures 3a–3b compare SPIKED-CFR with random projection (RP) and PCA across $k \in \{8, 16, 32, 64, 128\}$; we report relative improvements so $\epsilon_{\text{PEHE}}$ and $\epsilon_{\text{ATE}}$ are comparable (absolute errors are in Appendix E.2). Across both semi-synthetic and real-world datasets, SPIKED-CFR consistently improves over RP and PCA, and the gains persist across a wide range of $k$, suggesting low sensitivity to the projection dimension. Moreover, the raincloud plots in Figures 3c–3d show reduced variance across shared random seeds on representative datasets, indicating that learning $U$ via WPP stabilizes matching rather than adding randomness; additional raincloud plots are in Appendix E.1. Overall, we can find that: unlike PCA/RP, SPIKED-CFR explicitly searches for the most imbalanced $k$-dimensional subspace, so balancing focuses on the "spike".

**Which LLM layer to extract representations.** Holding SPIKED-CFR fixed, Figure 4a shows that representations extracted from different layers lead to noticeably different causal errors, with L-4 consistently performing best (errors are normalized to make $\epsilon_{\text{PEHE}}$ and $\epsilon_{\text{ATE}}$ comparable; see

Appendix E.3 for unnormalized results). Figure 4b further shows that L-4 yields one of the largest treated–control Wasserstein discrepancies (WPP projected), aligning with its superior causal performance. This matches our motivation: layers that better expose treated–control distributional differences provide a cleaner balancing target, while late pooling (e.g., LTH, the last-token hidden state) can wash out this signal and reduce overlap. Therefore, we use L-4 throughout.

**Overlap and balance in representation space.** We verify that SPIKED-CFR improves overlap by reducing treatment separability in the learned representations. On ACIC post, we train a logistic regression classifier to predict treatment from representations and report both predicted propensity distributions (Figures 5a–5c) and AUC (Figure 5d). Raw LLM embeddings are highly separable (AUC close to 1) with nearly disjoint propensity mass, indicating severe non-overlap in representation space. CFR-Wass partially reduces separability, suggesting that full-space Wasserstein balancing can remain unreliable under finite samples. In contrast, SPIKED-CFR drives AUC close to random guessing and produces substantially overlapping propensity distributions,

directly indicating improved balance between treated and control representations. Overall, SPIKED-CFR targets the confounding "spike" via worst-case low-dimensional balancing while keeping full $\Phi(X)$ for outcome prediction.

## 6. Conclusion

We proposed SPIKED-CFR, which enables stable Wasserstein-style balancing for high-dimensional LLM embeddings via Wasserstein Projection Pursuit under a spiked structure. Theory shows the projected discrepancy admits rates governed by $k \ll D$, avoiding vacuous high-dimensional guarantees. Across eight post-based benchmarks, SPIKED-CFR achieves stronger and more robust effect estimation than prior baselines.

The main limitations suggest three directions for future work. First, our method targets a balancing-relevant low-dimensional discrepancy subspace, rather than recovering the true latent confounders; as with unconfoundedness, the spiked structure itself cannot be directly verified from observational data. Second, the framework depends on upstream LLM extraction, where missed pre-treatment confounders, residual treatment or outcome leakage, and representation distortions may still bias downstream estimates. Third, the current formulation focuses on binary treatments, and extending projected balancing to multi-valued, continuous, and time-varying treatment settings remains an important direction for future work.

## Acknowledgments

This work was supported in part by the Major Program of the National Natural Science Foundation of China under Grant 72192823, and by the National Natural Science Foundation of China under Grant 62125206.

## Impact Statement

The development and future adoption of SPIKED-CFR as a framework for causal representation learning from high-dimensional LLM embeddings may benefit causal inference and AI-driven scientific discovery. By mitigating high-dimensional Wasserstein estimation challenges in semantic spaces, our work supports more robust treatment-effect estimation from unstructured data in domains such as healthcare and public policy. At the same time, estimates from observational text may be affected by hidden confounding, extraction errors, privacy risks, and dataset biases. Such methods should therefore be used as research tools for hypothesis generation and evaluation, rather than as standalone evidence for clinical or policy decisions.

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

# Appendix: SPIKED-CFR

## Table of Contents

## A. Additional Preliminaries and Notation

This appendix formalizes the loss functionals and discrepancy terms underlying Eq. (1) and our Wasserstein projection pursuit (WPP) objective.

### A.1. CFR risks and PEHE

**Potential outcomes and conditional means.** Let $\mu_t(x) := \mathbb{E}[Y(t) \mid X = x]$ for $t \in \{0, 1\}$ and $\tau(x) := \mu_1(x) - \mu_0(x)$. Given a representation $\Phi : \mathcal{X} \to \mathbb{R}^D$ and treatment-specific predictors $h_0, h_1$, we write $\hat{\mu}_t(x) := h_t(\Phi(x))$ and $\hat{\tau}(x) := \hat{\mu}_1(x) - \hat{\mu}_0(x)$.

**PEHE.** Following Hill (2011); Shalit et al. (2017), the Precision in Estimating Heterogeneous Effects (PEHE) is

$$\epsilon_{\text{PEHE}}(h, \Phi) := \mathbb{E}\left[\left(\hat{\tau}(X) - \tau(X)\right)^2\right] = \int_{\mathcal{X}} \left(\hat{\tau}(x) - \tau(x)\right)^2 dP_X(x). \tag{13}$$

**Factual and counterfactual risks.** Let $\ell : \mathbb{R} \times \mathbb{R} \to \mathbb{R}_+$ be the squared loss $\ell(\hat{y}, y) = (\hat{y} - y)^2$ in the theoretical development of Shalit et al. (2017). Denote $u := \mathbb{P}(T = 1)$ and $P_X^t := \mathrm{Law}(X \mid T = t)$. Define the *group-conditional factual risks*

$$\epsilon_F^t(h, \Phi) := \mathbb{E}[\ell(h_t(\Phi(X)), Y) \mid T = t], \qquad t \in \{0, 1\}, \tag{14}$$

and the (unweighted) factual risk used in the CFR bound,

$$\epsilon_F(h, \Phi) := \epsilon_F^0(h, \Phi) + \epsilon_F^1(h, \Phi). \tag{15}$$

Likewise, define the *group-conditional counterfactual risks*

$$\epsilon_{CF}^t(h, \Phi) := \mathbb{E}[\ell(h_t(\Phi(X)), Y(t)) \mid T = 1 - t], \qquad t \in \{0, 1\}, \tag{16}$$

and $\epsilon_{CF}(h, \Phi) := \epsilon_{CF}^0(h, \Phi) + \epsilon_{CF}^1(h, \Phi)$. Note that $\epsilon_F$ is observable, while $\epsilon_{CF}$ depends on unobserved counterfactuals.

**A variance constant.** Let $\sigma_{Y_t}^2(P)$ denote the conditional variance functional

$$\sigma_{Y_t}^2(P) := \mathbb{E}\Big[\big(Y(t) - \mu_t(X)\big)^2\Big], \qquad \sigma_Y^2 := \min\{\sigma_{Y_0}^2(P), \sigma_{Y_1}^2(P)\}. \tag{17}$$

As in Shalit et al. (2017), $\sigma_Y^2$ contributes only an additive constant in the PEHE bound.

## A.2. Integral probability metrics and the CFR bound

**Integral probability metrics.** Let $\mathcal{G}$ be a family of measurable functions $g : \mathcal{S} \to \mathbb{R}$. For probability measures $P, Q$ on $\mathcal{S}$, the induced integral probability metric (IPM, (Sriperumbudur et al., 2012; Müller, 1997)) is

$$\mathrm{IPM}_\mathcal{G}(P, Q) := \sup_{g \in \mathcal{G}} |\mathbb{E}_{Z \sim P}[g(Z)] - \mathbb{E}_{Z \sim Q}[g(Z)]| . \tag{18}$$

When $\mathcal{G}$ is symmetric ($g \in \mathcal{G} \Rightarrow -g \in \mathcal{G}$), the absolute value can be dropped.

**Wasserstein as an IPM and the role of $p$.** For $\mathcal{S} = \mathbb{R}^D$ with Euclidean metric, the Kantorovich–Rubinstein duality gives $\mathrm{IPM}_{\mathrm{Lip}(1)}(P, Q) = W_1(P, Q)$, where $\mathrm{Lip}(1)$ is the set of 1-Lipschitz functions (Villani et al., 2008). For $p > 1$, $W_p$ is not itself an IPM in this simple form, but it *dominates* $W_1$: for any $p \geq 1$ and measures with finite $p$-th moment, $W_1(P, Q) \leq W_p(P, Q)$ (Villani et al., 2008). Thus, any PEHE upper bound expressed in terms of $W_1$ remains valid (possibly looser) if we replace $W_1$ by $W_p$. This observation justifies the notation $\mathrm{IPM}(P_\Phi^1, P_\Phi^0) = W_p(P_\Phi^1, P_\Phi^0)$ adopted in the main text.

**Representation regularity and hypothesis control.** The CFR analysis of Shalit et al. (2017) requires that the loss-composed hypothesis belongs to the IPM class after transporting it through the inverse representation. We record this as a standing assumption.

**Assumption A.1** (Invertible representation and bounded loss-composition). The representation $\Phi : \mathcal{X} \to \mathbb{R}^D$ is one-to-one on $\mathrm{supp}(P_X)$ with measurable inverse $\Psi$. Moreover, there exists $B_\Phi > 0$ and a symmetric function class $\mathcal{G}$ such that for each $t \in \{0, 1\}$, the function

$$g_{\Phi, h, t}(r) := \frac{1}{B_\Phi} \mathbb{E}[\ell(h_t(r), Y(t)) \mid X = \Psi(r)] \quad \text{satisfies} \quad g_{\Phi, h, t} \in \mathcal{G}. \tag{19}$$

**Induced treated/control measures.** Let $P_\Phi^t := \mathrm{Law}(\Phi(X) \mid T = t)$ be the pushed distributions in representation space. In the notation of the main text, this is exactly $P_\Phi^t$.

**A standard CFR generalization bound.** Under Assumption A.1 and strong ignorability (Sec. 3), Shalit et al. (2017) show that $\epsilon_{CF}$ can be controlled by $\epsilon_F$ plus an IPM between $P_\Phi^1$ and $P_\Phi^0$, and then relate $\epsilon_{\mathrm{PEHE}}$ to $\epsilon_F + \epsilon_{CF}$. Specializing their result to squared loss yields (up to the same constant terms as in Shalit et al. (2017)):

$$\epsilon_{\mathrm{PEHE}}(h, \Phi) \leq 2\,\epsilon_F^0(h, \Phi) + 2\,\epsilon_F^1(h, \Phi) + 2B_\Phi\,\mathrm{IPM}_\mathcal{G}(P_\Phi^1, P_\Phi^0) - 4\sigma_Y^2. \tag{20}$$

Setting $\epsilon_F(h, \Phi) := 2(\epsilon_F^0 + \epsilon_F^1)$, $\alpha := 2B_\Phi$ and $\mathrm{Const} := -4\sigma_Y^2$ recovers the form in Eq. (1). When $\mathcal{G} = \mathrm{Lip}(1)$, $\mathrm{IPM}_\mathcal{G} = W_1 \leq W_p$, which yields the Wasserstein instantiation used in the main text.

## A.3. Wasserstein distances, projections, and empirical measures

**$p$-Wasserstein distance.**   For probability measures $P, Q$ on $\mathbb{R}^D$ with finite $p$-th moments (Villani et al., 2008),

$$W_p(P,Q) := \left( \inf_{\pi \in \Pi(P,Q)} \int_{\mathbb{R}^D \times \mathbb{R}^D} \|x - y\|_2^p \, d\pi(x,y) \right)^{1/p}, \tag{21}$$

where $\Pi(P,Q)$ is the set of couplings with marginals $P$ and $Q$.

**Push-forward under linear maps.**   For a measurable map $A : \mathbb{R}^D \to \mathbb{R}^k$, define the push-forward $A^\sharp P$ by $(A^\sharp P)(B) = P(A^{-1}(B))$ for any measurable $B \subset \mathbb{R}^k$. In particular, $U^\sharp P$ denotes the distribution of $UZ$ when $Z \sim P$.

**Stiefel manifold for row-orthonormal projections.**   We use the row-orthonormal Stiefel manifold (Absil et al., 2008)

$$\mathcal{V}_k := \{U \in \mathbb{R}^{k \times D} : UU^\top = I_k\}. \tag{22}$$

For $U \in \mathcal{V}_k$, the operator norm satisfies $\|U\|_{\mathrm{op}} = 1$, hence the map $z \mapsto Uz$ is 1-Lipschitz.

**Contraction under projection.**   Because $z \mapsto Uz$ is 1-Lipschitz, Wasserstein distances contract: for any $U \in \mathcal{V}_k$ and any $P, Q$ on $\mathbb{R}^D$ with finite $p$-moments,

$$W_p\big(U^\sharp P, U^\sharp Q\big) \leq W_p(P, Q). \tag{23}$$

**Projection-robust Wasserstein (WPP/PRW).**   We recall Eq. (7) (Paty & Cuturi, 2019; Niles-Weed & Rigollet, 2022; Lin et al., 2020):

$$\mathcal{W}_{p,k}(P,Q) := \max_{U \in \mathcal{V}_k} W_p\big(U^\sharp P, U^\sharp Q\big). \tag{24}$$

**Empirical measures and estimators.**   Given i.i.d. samples $\{Z_i\}_{i=1}^n$ from $P$, let $\hat{P}_n := \frac{1}{n} \sum_{i=1}^n \delta_{Z_i}$. In our causal setting, for $t \in \{0,1\}$ let $I_t := \{i : T_i = t\}$, $n_t := |I_t|$, and

$$\hat{P}_\Phi^t := \frac{1}{n_t} \sum_{i \in I_t} \delta_{\Phi(X_i)}. \tag{25}$$

Define $\widehat{W}_p := W_p(\hat{P}_\Phi^1, \hat{P}_\Phi^0)$ and the empirical WPP estimator

$$\widehat{\mathcal{W}}_{p,k} := \mathcal{W}_{p,k}(\hat{P}_\Phi^1, \hat{P}_\Phi^0) = \max_{U \in \mathcal{V}_k} W_p\Big(U^\sharp \hat{P}_\Phi^1, U^\sharp \hat{P}_\Phi^0\Big). \tag{26}$$

## A.4. Spiked structure and statistical regularity for WPP

**Spiked structure.**   Assumption 4.3 in Sec. 4.3 is the causal analogue of the "spiked transport model" (Niles-Weed & Rigollet, 2022): the treated/control shift lies in a $k$-dimensional subspace $\mathcal{S}$ and the orthogonal component is shared. We additionally record two standard regularity conditions used in the statistical analysis of WPP.

**Assumption A.2** (Dimension regime). We work in a regime where $D > 2p$ and $k > 2p$, and $P_\Phi^0, P_\Phi^1$ have finite $p$-th moments.

**Assumption A.3** (Transport inequality for concentration). For $t \in \{0,1\}$, the measures $P_\Phi^t$ satisfy a $T_p(\sigma^2)$ transport-entropy inequality (Gozlan & Léonard, 2010): for all probability measures $\nu \ll P_\Phi^t$,

$$W_p^2(\nu, P_\Phi^t) \leq 2\sigma^2 \, \mathrm{KL}(\nu \| P_\Phi^t). \tag{27}$$

**Discrepancy preservation under the spike.**   Under Assumption 4.3, Wasserstein discrepancy is entirely carried by the spiked subspace.

**Lemma A.4** (Wasserstein depends only on the spike under Assumption 4.3). *Let $P_\Phi^t = \mathrm{Law}(S^{(t)} + \varepsilon)$ with $S^{(t)} \in \mathcal{S}$ and $\varepsilon \in \mathcal{S}^\perp$ satisfying $\mathrm{Law}(\varepsilon \mid T = 1) = \mathrm{Law}(\varepsilon \mid T = 0)$ as in Assumption 4.3. Let $U_\star \in \mathcal{V}_k$ have row span $\mathcal{S}$. Then for any $p \geq 1$,*

$$W_p(P_\Phi^1, P_\Phi^0) = W_p\big(U_\star^\sharp P_\Phi^1, \ U_\star^\sharp P_\Phi^0\big) = W_p\Big(\mathrm{Law}(U_\star S^{(1)}), \ \mathrm{Law}(U_\star S^{(0)})\Big), \tag{28}$$

*and in particular $\mathcal{W}_{p,k}(P_\Phi^1, P_\Phi^0) = W_p(P_\Phi^1, P_\Phi^0)$.*

*Proof.* Because $U_\star$ annihilates $\mathcal{S}^\perp$, we have $U_\star(S^{(t)} + \varepsilon) = U_\star S^{(t)}$. Hence $U_\star^\sharp P_\Phi^t = \mathrm{Law}(U_\star S^{(t)})$ for $t \in \{0, 1\}$. By contraction (23), $W_p(U_\star^\sharp P_\Phi^1, U_\star^\sharp P_\Phi^0) \leq W_p(P_\Phi^1, P_\Phi^0)$.

For the reverse inequality, consider any coupling $\pi$ of $\mathrm{Law}(S^{(1)})$ and $\mathrm{Law}(S^{(0)})$. Couple the shared noise by taking $\varepsilon$ identical in both groups, i.e., use the product coupling $\tilde{\pi}$ of $(S^{(1)} + \varepsilon)$ and $(S^{(0)} + \varepsilon)$ induced by $\pi$ and $\varepsilon$. Then $\|(S^{(1)} + \varepsilon) - (S^{(0)} + \varepsilon)\|_2 = \|S^{(1)} - S^{(0)}\|_2$, so taking infima over couplings yields $W_p(P_\Phi^1, P_\Phi^0) \leq W_p(\mathrm{Law}(S^{(1)}), \mathrm{Law}(S^{(0)})) = W_p(U_\star^\sharp P_\Phi^1, U_\star^\sharp P_\Phi^0)$. Finally, $\mathcal{W}_{p,k} \geq W_p(U_\star^\sharp P_\Phi^1, U_\star^\sharp P_\Phi^0)$ by definition, and $\mathcal{W}_{p,k} \leq W_p(P_\Phi^1, P_\Phi^0)$ by (23), proving equality. $\square$

## B. Proofs & Theoretical Analysis

### B.1. Regularity conditions (recall)

This section collects the technical assumptions used by Proposition 4.4 and Theorem 4.5. We do not introduce new assumptions beyond Appendix A: (i) the CFR/IPM regularity in Assumption A.1, (ii) the spiked structure in Assumption 4.3, and (iii) the statistical conditions for WPP estimation in Assumptions A.2–A.3. All proofs below only invoke these assumptions.

### B.2. Derivation of Eq. (4) and Corollary 4.2

We prove the bound claimed in Corollary 4.2.

**Step 1: from population discrepancy to empirical discrepancy.** Instantiate Eq. (1) with Wasserstein and write

$$\epsilon_{\mathrm{PEHE}}(h, \Phi) \;\leq\; \epsilon_F(h, \Phi) + \alpha\, W_p(P_\Phi^1, P_\Phi^0) + \mathrm{Const}. \tag{29}$$

For any scalars $a, b$, $a \leq b + |a - b|$; applying this to $a = W_p(P_\Phi^1, P_\Phi^0)$ and $b = W_p(\hat{P}_\Phi^1, \hat{P}_\Phi^0) = \widehat{W}_p$ yields

$$W_p(P_\Phi^1, P_\Phi^0) \;\leq\; \widehat{W}_p + \left|W_p(\hat{P}_\Phi^1, \hat{P}_\Phi^0) - W_p(P_\Phi^1, P_\Phi^0)\right| = \widehat{W}_p + \Delta_{n,D}, \tag{30}$$

where $\Delta_{n,D}$ is exactly as defined in Eq. (3). Substituting gives

$$\epsilon_{\mathrm{PEHE}}(h, \Phi) \;\leq\; \epsilon_F(h, \Phi) + \alpha\, \widehat{W}_p + \alpha\, \Delta_{n,D} + \mathrm{Const}, \tag{31}$$

which is Eq. (4).

**Step 2: a lower bound for the estimation error.** Proposition 4.1 states that when $D > 2p$, the worst-case expected estimation error of empirical Wasserstein distances satisfies $\mathbb{E}[\Delta_{n,D}] \gtrsim n^{-1/D}$, which is Eq. (5). Combining Step 1 and Step 2 yields Corollary 4.2. $\square$

### B.3. Proof of Proposition 4.4 (WPP estimation rate)

We provide a proof sketch with precise source mapping, since Proposition 4.4 is an immediate specialization of the main upper bound for the WPP estimator in the spiked transport model (Niles-Weed & Rigollet, 2022).

**Mapping to Niles-Weed & Rigollet (2022).** Identify their ambient dimension $d$ with our $D$, their spike dimension with our $k$, their pair of measures $(\mu^{(1)}, \mu^{(2)})$ with $(P_\Phi^1, P_\Phi^0)$, and their WPP estimator $\widehat{W}_{p,k}$ with our $\widehat{\mathcal{W}}_{p,k}$. Our Assumption 4.3 matches their spiked transport model specialization where the orthogonal component is shared; Assumption A.3 matches their $T_p(\sigma^2)$ condition.

**Rate statement.** Under Assumptions 4.3 and A.3, Niles-Weed & Rigollet (2022, Theorem 1) show that

$$\mathbb{E}\left[\left|\widehat{\mathcal{W}}_{p,k} - W_p(P_\Phi^1, P_\Phi^0)\right|\right] \;\lesssim\; r_{p,k}(n) \;+\; \sigma\sqrt{\frac{D \log n}{n}}, \tag{32}$$

where $n = \min\{n_0, n_1\}$ and $r_{p,k}(n)$ is the canonical $k$-dimensional Wasserstein estimation rate. In the regime $k > 2p$ (Assumption A.2), $r_{p,k}(n) \asymp n^{-1/k}$ (up to constants depending on $p$ and the $T_p(\sigma^2)$ parameter). This yields

$$\mathbb{E}\left[\left|\widehat{\mathcal{W}}_{p,k} - W_p(P_\Phi^1, P_\Phi^0)\right|\right] \;\lesssim\; n^{-1/k} \;+\; \sqrt{\frac{D \log n}{n}}. \tag{33}$$

Finally, Lemma A.4 implies $W_p(P_\Phi^1, P_\Phi^0) = \mathcal{W}_{p,k}(P_\Phi^1, P_\Phi^0)$, so Eq. (32) is equivalent to Eq. (8). $\qquad\square$

### B.4. Proof of Theorem 4.5 (Spiked-CFR generalization bound)

We prove Eq. (9).

**Step 1: start from the CFR template.** By Eq. (1) (a direct consequence of Shalit et al. (2017); see Appendix A),

$$\epsilon_{\text{PEHE}}(\hat{h}, \hat{\Phi}) \;\le\; \epsilon_F(\hat{h}, \hat{\Phi}) + \alpha\, W_p(P_{\hat{\Phi}}^1, P_{\hat{\Phi}}^0) + \text{Const.} \tag{34}$$

**Step 2: replace $W_p$ by the projection-robust discrepancy.** Under Assumption 4.3, Lemma A.4 gives $W_p(P_{\hat{\Phi}}^1, P_{\hat{\Phi}}^0) = \mathcal{W}_{p,k}(P_{\hat{\Phi}}^1, P_{\hat{\Phi}}^0)$. Hence

$$\epsilon_{\text{PEHE}}(\hat{h}, \hat{\Phi}) \;\le\; \epsilon_F(\hat{h}, \hat{\Phi}) + \alpha\, \mathcal{W}_{p,k}(P_{\hat{\Phi}}^1, P_{\hat{\Phi}}^0) + \text{Const.} \tag{35}$$

**Step 3: population-to-empirical decomposition for WPP.** Applying the inequality $a \le b + |a - b|$ with $a = \mathcal{W}_{p,k}(P_{\hat{\Phi}}^1, P_{\hat{\Phi}}^0)$ and $b = \widehat{\mathcal{W}}_{p,k} = \mathcal{W}_{p,k}(\hat{P}_{\hat{\Phi}}^1, \hat{P}_{\hat{\Phi}}^0)$ yields

$$\mathcal{W}_{p,k}(P_{\hat{\Phi}}^1, P_{\hat{\Phi}}^0) \;\le\; \widehat{\mathcal{W}}_{p,k} + \left| \widehat{\mathcal{W}}_{p,k} - \mathcal{W}_{p,k}(P_{\hat{\Phi}}^1, P_{\hat{\Phi}}^0) \right|. \tag{36}$$

Substituting into (34),

$$\epsilon_{\text{PEHE}}(\hat{h}, \hat{\Phi}) \;\le\; \epsilon_F(\hat{h}, \hat{\Phi}) + \alpha\, \widehat{\mathcal{W}}_{p,k} + \alpha \left| \widehat{\mathcal{W}}_{p,k} - \mathcal{W}_{p,k}(P_{\hat{\Phi}}^1, P_{\hat{\Phi}}^0) \right| + \text{Const.} \tag{37}$$

**Step 4: apply the WPP estimation rate.** Taking expectations in (37) and applying Proposition 4.4 yields

$$\mathbb{E}\big[\epsilon_{\text{PEHE}}(\hat{h}, \hat{\Phi})\big] \;\le\; \mathbb{E}\big[\epsilon_F(\hat{h}, \hat{\Phi})\big] + \alpha\, \mathbb{E}\big[\widehat{\mathcal{W}}_{p,k}\big] + \mathcal{O}\left( n^{-1/k} + \sqrt{\frac{D \log n}{n}} \right) + \text{Const,} \tag{38}$$

where the big-$\mathcal{O}$ absorbs $\alpha$ and universal constants. This is Eq. (9). $\qquad\square$

## C. Optimization on the Stiefel Manifold

This appendix provides optimization details for the min–max objective in Eq. (10), focusing on the adversarial update of the projection matrix $U \in \mathcal{V}_k = \{U \in \mathbb{R}^{k \times D} : UU^\top = I_k\}$. Algorithm 1 summarizes the overall alternating training procedure.

### C.1. Geometry of $\mathcal{V}_k$ with row-orthonormal constraints

**Tangent space.** Let $U \in \mathcal{V}_k$. Differentiating the constraint $UU^\top = I_k$ gives $\Delta U^\top + U\Delta^\top = 0$. Hence the tangent space (Absil et al., 2008; Edelman et al., 1998) is

$$T_U \mathcal{V}_k = \left\{ \Delta \in \mathbb{R}^{k \times D} : \Delta U^\top + U\Delta^\top = 0 \right\}. \tag{39}$$

**Orthogonal projection onto the tangent space.** Given an ambient (Euclidean) matrix $G \in \mathbb{R}^{k \times D}$, its orthogonal projection onto $T_U \mathcal{V}_k$ under the Frobenius inner product (Absil et al., 2008; Edelman et al., 1998) is

$$\Pi_U(G) = G - \text{sym}(GU^\top)U, \qquad \text{sym}(A) := \tfrac{1}{2}(A + A^\top). \tag{40}$$

One checks that $\Pi_U(G) \in T_U \mathcal{V}_k$ since $\Pi_U(G)U^\top + U\Pi_U(G)^\top = GU^\top + UG^\top - 2\,\text{sym}(GU^\top) = 0$.

**Retractions.** A retraction $\text{Retr}_U : T_U \mathcal{V}_k \to \mathcal{V}_k$ maps a tangent step back to the manifold. Two standard choices for the row-orthonormal Stiefel (Absil et al., 2008; Edelman et al., 1998) are:

---

**Algorithm 1** SPIKED-CFR Training Procedure

---

1: **Input:** Raw posts $\mathcal{D} = \{W_i\}_{i=1}^n$, subspace dim $k$, hyperparams $\alpha$, adversary steps $\mathcal{M}_{\text{adv}}$.
2: **Preprocessing:** Extract tuples $(T_i, Y_i, \tilde{X}_i)$ via LLM prompting.
3: **Frozen encoding:** Compute $X_i = f_\theta(\tilde{X}_i)$ for all $i$ .
4: **Initialize:** Adapter $\Phi$, heads $h_0, h_1$, and $U \in \mathcal{V}_k$.
5: **for** epoch $= 1$ **to** $N_{\text{epochs}}$ **do**
6:     **for** minibatch $\mathcal{B} \subset \{(X_i, T_i, Y_i)\}$ **do**
7:         Compute representations $\Phi(X_i)$ for $i \in \mathcal{B}$.
8:         Split $\mathcal{B}$ into $\mathcal{B}_1 = \{i \in \mathcal{B} : T_i = 1\}$ and $\mathcal{B}_0 = \{i \in \mathcal{B} : T_i = 0\}$.
9:         // Adversary ascent (maximize discrepancy)
10:         **for** $m = 1$ **to** $\mathcal{M}_{\text{adv}}$ **do**
11:            Compute $\nabla_U W_p\left(U^\sharp \hat{P}^1_{\Phi,\mathcal{B}}, U^\sharp \hat{P}^0_{\Phi,\mathcal{B}}\right)$.
12:            Update $U$ via Riemannian gradient ascent.
13:         **end for**
14:         // Learner descent (minimize prediction + balance)
15:         Compute $\mathcal{L} = \frac{1}{|\mathcal{B}|} \sum_{i \in \mathcal{B}} \ell(h_{T_i}(\Phi(X_i)), Y_i) + \alpha\, W_p\left(U^\sharp \hat{P}^1_{\Phi,\mathcal{B}}, U^\sharp \hat{P}^0_{\Phi,\mathcal{B}}\right)$.
16:         Update $\Phi, h_0, h_1 \leftarrow \text{Optimizer}(\nabla_{\Phi, h_0, h_1} \mathcal{L})$.
17:     **end for**
18: **end for**
19: **Output:** Trained adapter $\Phi$ and heads $h_0, h_1$.

---

- **Polar retraction:**
$$\text{Retr}_U(\tilde{U}) := (\tilde{U}\tilde{U}^\top)^{-1/2}\tilde{U}, \qquad \tilde{U} \in \mathbb{R}^{k \times D} \text{ with } \tilde{U}\tilde{U}^\top \succ 0, \tag{41}$$
which enforces $(\text{Retr}_U(\tilde{U}))(\text{Retr}_U(\tilde{U}))^\top = I_k$.

- **QR retraction (implemented via transpose):** compute a thin QR factorization of $\tilde{U}^\top = QR$ with $Q \in \mathbb{R}^{D \times k}$ having orthonormal columns, then set
$$\text{Retr}_U(\tilde{U}) := Q^\top. \tag{42}$$

### C.2. Riemannian ascent for the adversary $U$

**Adversary objective.** For a minibatch $\mathcal{B}$, define empirical treated/control measures on $\mathbb{R}^D$ by $\hat{P}^t_{\Phi,\mathcal{B}} = \frac{1}{|\mathcal{B}_t|} \sum_{i \in \mathcal{B}_t} \delta_{\Phi(X_i)}$. The adversary maximizes
$$F(U) := W_p\left(U^\sharp \hat{P}^1_{\Phi,\mathcal{B}}, U^\sharp \hat{P}^0_{\Phi,\mathcal{B}}\right), \qquad U \in \mathcal{V}_k. \tag{43}$$

In practice we use a differentiable entropic-regularized Wasserstein (Sinkhorn) approximation (Liao et al.; Yue et al., 2026), so $F$ is smooth and $\nabla_U F$ is obtained via automatic differentiation.

**Riemannian gradient.** Let $G := \nabla_U F(U)$ be the Euclidean gradient. The Riemannian gradient on $\mathcal{V}_k$ under the canonical metric is the tangent projection
$$\text{grad}\, F(U) = \Pi_U(G) = G - \text{sym}(GU^\top)\, U, \tag{44}$$
using (40).

**Update rule.** Given step size $\eta > 0$, a single Riemannian ascent step is
$$\tilde{U} \leftarrow U + \eta\, \text{grad}\, F(U), \qquad U \leftarrow \text{Retr}_U(\tilde{U}), \tag{45}$$
where $\text{Retr}_U$ is either (41) or (42). This is the "Riemannian gradient ascent" operation referenced in Algorithm 1.

### C.3. Differentiable Wasserstein via Sinkhorn

**Entropic OT on projected features.** Let $\{z_i\}_{i \in \mathcal{B}_1}$ and $\{z'_j\}_{j \in \mathcal{B}_0}$ denote the minibatch representations in $\mathbb{R}^D$. Given $U \in \mathcal{V}_k$, define projected points $\tilde{z}_i := Uz_i \in \mathbb{R}^k$ and $\tilde{z}'_j := Uz'_j$. We form the cost matrix $C_{ij} := \|\tilde{z}_i - \tilde{z}'_j\|_2^p$ and compute

the entropic OT value (Peyré & Cuturi, 2019; Cuturi, 2013; Genevay et al., 2018)

$$W_{p,\varepsilon}^p := \min_{\pi \in \Pi(a,b)} \langle \pi, C \rangle + \varepsilon \sum_{i,j} \pi_{ij}(\log \pi_{ij} - 1), \tag{46}$$

where $a, b$ are uniform weights on $\mathcal{B}_1, \mathcal{B}_0$ and $\varepsilon > 0$ is the Sinkhorn regularization. We set $F(U) := W_{p,\varepsilon}(U^\sharp \hat{P}_{\Phi,\mathcal{B}}^1, U^\sharp \hat{P}_{\Phi,\mathcal{B}}^0)$. Because $C$ depends smoothly on $U$, and the Sinkhorn iterations are differentiable (Peyré & Cuturi, 2019; Liao et al., 2024a;b), $F$ admits stable gradients in practice.

**Interaction with the learner.** During the adversary step, $(\Phi, h_0, h_1)$ are held fixed and we ascend in $U$. During the learner step, $U$ is held fixed and we descend in $(\Phi, h_0, h_1)$ on $\mathcal{L}_{\text{pred}} + \alpha \mathcal{L}_{\text{WPP}}$ (Sec. 4.4), backpropagating through the same Sinkhorn computation.

### C.4. Computational complexity

We summarize the per-minibatch computational cost of Algorithm 1. Let $|\mathcal{B}_1| = b_1, |\mathcal{B}_0| = b_0$, and denote $b := \max\{b_0, b_1\}$. Let $I$ be the number of Sinkhorn iterations used to approximate $W_p$ (Sec. C.3), and let $\mathcal{M}_{\text{adv}}$ be the number of adversary ascent steps per minibatch.

**Adversary ascent (one step).** Given current representations $z_i = \Phi(X_i) \in \mathbb{R}^D$, the projected features $\tilde{z}_i = U z_i \in \mathbb{R}^k$ cost

$$\mathcal{O}\big((b_0 + b_1)\, kD\big) = \mathcal{O}(bkD). \tag{47}$$

Constructing the cost matrix $C_{ij} = \|\tilde{z}_i - \tilde{z}_j'\|_2^p$ costs

$$\mathcal{O}(b_0 b_1\, k), \tag{48}$$

and dense Sinkhorn updates cost $\mathcal{O}(I\, b_0 b_1)$. The Riemannian projection $\Pi_U(G) = G - \text{sym}(GU^\top)U$ involves forming $GU^\top \in \mathbb{R}^{k \times k}$ and multiplying back by $U$, which costs $\mathcal{O}(k^2 D)$. A QR-based retraction on $\hat{U}^\top \in \mathbb{R}^{D \times k}$ costs $\mathcal{O}(Dk^2)$. Thus, one adversary step costs

$$\mathcal{O}\big(bkD + b_0 b_1\, k + I\, b_0 b_1 + Dk^2\big). \tag{49}$$

Over $\mathcal{M}_{\text{adv}}$ ascent steps, multiply the above by $\mathcal{M}_{\text{adv}}$.

**Learner descent (one step).** The learner update backpropagates through the factual loss and the Sinkhorn objective with $U$ fixed. Let $\mathsf{C}_\Phi$ denote the per-sample cost of the adapter/heads forward–backward pass (typically $\mathsf{C}_\Phi \ll$ the cost of running the frozen LLM). Then the learner step costs

$$\mathcal{O}\big(b\, \mathsf{C}_\Phi + bkD + b_0 b_1\, k + I\, b_0 b_1\big), \tag{50}$$

where the last three terms arise from forming the projected OT objective and differentiating through Sinkhorn.

**Memory.** Storing the dense cost/kernel matrices requires $\mathcal{O}(b_0 b_1)$ memory.

**Why projection helps.** Crucially, the OT cost construction scales with the *projection dimension* $k$ rather than the ambient dimension $D$, since distances are computed in $\mathbb{R}^k$ after applying $U$. Moreover, because $f_\theta$ is frozen, we can precompute $X_i = f_\theta(\tilde{X}_i)$ once and train only $(\Phi, h_0, h_1, U)$, substantially reducing end-to-end training cost in practice.

## D. Additional Experimental Details

### D.1. Datasets

**Dataset statistics.** Table 3 summarizes the benchmarks. Notably, "$N$" means the total sample size of each dataset, "Unit-level CF" indicates the availability of individual-level counterfactual ground truths. "GT ATE (Source)" reports the ground-truth ATE derived from underlying simulations or external RCTs.

*Table 3.* Summary of key statistics for all eight datasets.

| Dataset | Type | N | Treatment | Outcome | Outcome Type | Unit-level CF | GT ATE |
|---|---|---|---|---|---|---|---|
| IHDP | Semi-synthetic | 747 | Home Visit | IQ Score | Continuous | Yes | 4.19 (Hill, 2011) |
| ACIC | Semi-synthetic | 4,802 | Simulation | Synthetic $Y$ | Continuous | Yes | 3.47 (Dorie et al., 2019) |
| Hillstrom | Semi-synthetic | 2000 | Email communication | Website visit | Binary | No | 6.09 (Hillstrom, 2008) |
| Retail Hero | Semi-synthetic | 2000 | SMS communication | Purchase | Binary | No | 3.32 (Group, 2019) |
| Sema vs. Tirz | Real-world | 5000 | Corresponding drug | Weight loss $\geq$ 5% | Binary | No | 10.11 (Frías et al., 2021) |
| Sema vs. Lira | Real-world | 6191 | Corresponding drug | Weight loss $\geq$ 10% | Binary | No | -14.7 (Capehorn et al., 2020) |
| Eren vs. Topi | Real-world | 10000 | Corresponding drug | Discontinuation due to adverse effects | Binary | No | 28.30 (Reuter et al., 2022) |
| Onab vs. Topi | Real-world | 4788 | Corresponding drug | Discontinuation due to adverse effects | Binary | No | 41.00 (Rothrock et al., 2019) |

**IHDP-post/ACIC-post generation and extraction.** To obtain post-based versions of IHDP and ACIC with counterfactual supervision, we use GPT-4o-mini to (i) render each tabular instance into a first-person post and (ii) extract $(T, Y, \tilde{X})$ from the generated post via prompting, where $\tilde{X}$ denotes the pre-treatment text used as input to the frozen LLM encoder (Sec. 4.1). During post generation, the LLM is conditioned only on covariates, treatment, and the factual outcome; the counterfactual outcome is withheld and used solely for evaluation to avoid potential-outcome leakage into text. All prompt templates are provided in Appendix D.5. Beyond the benchmarks considered here, text-based causal effect estimation may also be useful in domains where interventions and outcomes are entangled with unstructured user behavior, such as recommendation systems (Liu et al., 2025b; 2026) and model safety or unlearning (Wang et al., 2025a; 2026). Systematic evaluation in these broader settings is left for future work.

## D.2. Additional Baseline Descriptions

We provide additional descriptions for the baselines used in §5. **BOW-OLS** fits a linear outcome model on bag-of-words features (CountVectorizer) as a shallow text baseline. **BERT-PSM** computes propensity scores and performs propensity-score matching (Rosenbaum & Rubin, 1983) using frozen BERT representations (Devlin et al., 2019). **BERT-OForest** estimates heterogeneous treatment effects on BERT embeddings using Orthogonal Random Forest (Oprescu et al., 2019). **BERT-CForest** estimates heterogeneous treatment effects on BERT embeddings using Causal Forest (Athey et al., 2019). **Sentence Encoder** encodes each post with a pretrained sentence embedding model (SBERT) (Reimers & Gurevych, 2019), predicts structured variables $(X, T, Y)$ from the post, and then applies a structured causal estimator to the predicted variables. For frozen-LLM baselines, **Llama-CForest** applies Causal Forest (Athey et al., 2019) on frozen Llama embeddings (Touvron et al., 2023). **Llama-TARNet** applies TARNet (Shalit et al., 2017) on frozen Llama embeddings (Touvron et al., 2023). **Llama-Dragonnet** applies Dragonnet (Shi et al., 2019) on frozen Llama embeddings (Touvron et al., 2023). **Llama-CFR-MMD** and **Llama-CFR-Wass** implement counterfactual regression (Shalit et al., 2017) with MMD and Wasserstein IPM regularization, respectively, on frozen Llama embeddings (Touvron et al., 2023). **CausalBERT** follows the text-adjustment approach in (Pryzant et al., 2021) for effect estimation from text. **NATURAL** is an end-to-end LLM baseline operating directly on posts as described in (Dhawan et al., 2024). Finally, **CFR-Wass (Structured)** runs CFR-Wass (Shalit et al., 2017) on the original tabular covariates as a non-text reference.

## D.3. Evaluation Metrics

We follow the potential outcomes notation in §3. Recall that the CATE is $\tau(x) = \mathbb{E}[Y(1) - Y(0) \mid X = x]$ and our estimator is $\hat{\tau}(X) = \hat{\mu}_1(X) - \hat{\mu}_0(X)$ with $\hat{\mu}_t(X) = h_t(\Phi(X))$.

**CATE error (PEHE).** On benchmarks that provide unit-level potential outcomes (IHDP-post and ACIC-post), we compute the *precision in estimation of heterogeneous effects* (PEHE, (Hill, 2011)) as the root mean squared error between the estimated effect and the ground-truth unit effect $\tau_i := Y_i(1) - Y_i(0)$:

$$\epsilon_{\text{PEHE}} = \sqrt{\frac{1}{n} \sum_{i=1}^{n} (\hat{\tau}(X_i) - \tau_i)^2}. \tag{51}$$

Lower $\epsilon_{\text{PEHE}}$ indicates more accurate heterogeneous effect estimation.

**ATE estimation and error.** For any test set of size $n$, we compute the estimated ATE by averaging predicted effects:

$$\widehat{\text{ATE}} = \frac{1}{n} \sum_{i=1}^{n} \hat{\tau}(X_i). \tag{52}$$

We report the absolute ATE error

$$\epsilon_{\text{ATE}} = \left| \widehat{\text{ATE}} - \text{ATE} \right|, \tag{53}$$

where ATE denotes the corresponding ground-truth average effect for the dataset.

**RMSE of ATE across runs.** Over $R$ independent runs, we report

$$\text{RMSE(ATE)} = \sqrt{\frac{1}{R} \sum_{r=1}^{R} \left( \widehat{\text{ATE}}^{(r)} - \text{ATE} \right)^2}, \tag{54}$$

where $\widehat{\text{ATE}}^{(r)}$ is the ATE estimate from run $r$.

### D.4. Reproducibility Details: Hyperparameters

By default, we use a two-layer adapter and outcome heads with hidden sizes $(1024, 1024)$ and $(60, 60)$, respectively, and set LR $= 10^{-3}$, batch size 128, and $\mathcal{M}_{\text{adv}} = 5$. We tune $\alpha$, $k$, dropout and weight decay per dataset using the same validation protocol and search space. Sensitivity analyses for $k$ and $\alpha$ are provided in Appendix E.4. Table 4 reports the hyperparameters of SPIKED-CFR on each dataset.

*Table 4.* Hyperparameters for different datasets.

| Dataset | Adapter dim | Outcome head dim | $\alpha$ | Dropout | LR | Weight decay | Batch size | Proj. dim $k$ | Adv. steps $\mathcal{M}_{\text{adv}}$ |
|---|---|---|---|---|---|---|---|---|---|
| IHDP | 1024, 1024 | 60, 60 | 1e+0 | 0 | 1e-3 | 1e-1 | 128 | 64 | 5 |
| ACIC | 1024, 1024 | 60, 60 | 1e+1 | 0.5 | 1e-3 | 1e-4 | 128 | 8 | 5 |
| Hillstorm | 1024, 1024 | 60, 60 | 1e+0 | 0.5 | 1e-3 | 1e-4 | 128 | 16 | 5 |
| Retail Hero | 1024, 1024 | 60, 60 | 1e-1 | 0.5 | 1e-3 | 1e-4 | 128 | 32 | 5 |
| Sema vs. Tirz | 1024, 1024 | 60, 60 | 1e-2 | 0.5 | 1e-3 | 1e-4 | 128 | 4 | 5 |
| Sema vs. Lira | 1024, 1024 | 60, 60 | 1e+0 | 0 | 1e-3 | 1e-3 | 128 | 10 | 5 |
| Eren vs. Topi | 1024, 1024 | 60, 60 | 1e-1 | 0 | 1e-3 | 1e-3 | 128 | 8 | 5 |
| Onab vs. Topi | 1024, 1024 | 60, 60 | 1e+0 | 0 | 1e-3 | 1e-4 | 128 | 16 | 5 |

### D.5. Prompts

**Post generation prompt (IHDP/ACIC).** For Hillstrom/RetailHero and the four Reddit clinical datasets, we use the original prompting and data curation pipeline from NATURAL (Dhawan et al., 2024); we only provide the prompts used to construct IHDP-post/ACIC-post here.

**IHDP-post generation prompt: parenting forum style**

You are an everyday mother participating in the Infant Health and Development Program (IHDP), a program for low-birth-weight premature infants. You are writing a post for a social media parenting group to share your personal journey.

## Your Persona
You are a mother who is very detail-oriented and keeps a strict diary of your child's progress in the IHDP program. You care deeply about the exact medical statistics because you are comparing progress with other families and want accurate advice. You are not a medical expert, and you are simply quoting numbers and checkboxes from your doctor's chart and the study paperwork. Even though you include lots of exact data, you still write in a casual, emotional social-media tone as an everyday mom. You may use short, simple sentences, but they must read as a flowing story, not a labeled list.

## Hard requirements
To ensure the post reflects the provided structured data:
• Use **ONLY** the information provided in ``Attributes'' (x_1--x_25, treatment, y_factual). Do **NOT** invent additional facts.
• **Continuous Variables**: You MUST include ALL continuous variables (x_1--x_6). Do not omit any. Frame standardized numbers (e.g., -0.53 or 1.2) as values you are reading directly from a medical report. Do not round them.
• **Binary Attributes**: All binary attributes x_7--x_25 must be expressed as clear natural-language facts with an **explicit** ``**Yes**'' or ``**No**'' for EACH attribute.
  -- Do NOT combine binary attributes into one statement.
  -- **Education (x_10--x_12)**: These are binary indicators for education. If all are ``No,'' interpret as ``College graduate.'' You must still state the ``No'' facts for each.
  -- **Site (x_19--x_25)**: These are binary indicators for site. If all are ``No,'' interpret as ``Site 8.'' You must still state the ``No'' facts for each.
• **Outcome**: Explicitly state whether you participated in the early intervention program (treatment), and mention your child's cognitive development score (y_factual) with the exact value.
• **Style**: Do not write in a checklist style. Write as a natural first-person social media post with normal paragraphs and transitions.

## Output requirements
Output **ONLY** the post body text. No title, no explanation, no extra formatting.

**ACIC-post generation prompt: clinical note style**

```
## Role
You are an experienced Attending Pediatrician conducting a 12-month developmental
follow-up assessment.

## Task
Synthesize the provided raw patient data into a professional **Clinical Medical Note**.
The data includes maternal history, pregnancy/delivery details, and the infant's
health status over the first year.

## Data Dictionary (Reference for Input Variables)
The following mapping defines the input variables:
 • Maternal Demographics & Social: x_1 (Age), x_2 (Marital Status), x_5 (Height),
   x_6 (Pre-preg Weight), x_17 (Work Status), x_18 (Years Educ), x_19 (Income), x_20
   (Housing Density), x_21 (Birth Place), x_22 (Consanguinity), x_23 (Socio-Econ
   Index), x_24 (Race), x_25 (Age Menarche), x_26 (Diastolic BP), x_27 (Mom Birth
   Weight).
 • Maternal Health & Habits: x_3 (Cigs/Day), x_4 (Years Smoked), x_7 (Cardio Cond),
   x_8 (Pulm Cond), x_9 (Hema Cond), x_10 (Endo Cond), x_11 (Veneral Cond), x_12 (Urin
   Cond), x_13 (Gyne Cond), x_14 (Neur Cond), x_15 (Obst Compl), x_16 (Infect Dis).
 • Paternal: x_28 (Dad Age), x_29 (Dad Years Educ).
 • Pregnancy History: x_30 (Prev Premes), x_31 (Prev Abortions), x_32 (Prev Pregs),
   x_33 (Prev Stillbirths).
 • Birth & Delivery: x_36 (Placental Weight), x_37 (Cord Length), x_38 (Infant Sex),
   x_39 (Apgar 1m), x_40 (Apgar 5m), x_58 (Gestational Age), treatment (Birth Weight
   Category).
 • Infant Health & Labs: x_41 (Bottle Days), x_42 (Breast Days), x_43 (Bilirubin),
   x_44 (Hematocrit), x_45 (Hemoglobin), x_46 (Neur Abn), x_47 (CNS Cond), x_48
   (Muscoskel), x_49 (Resp Abn), x_50 (Cardio Abn), x_51 (Liver Abn), x_52 (Hema Cond),
   x_53 (Infect Dis), x_54 (Syndrome), x_55 (Endo Dis), x_56 (Medical Procs).
 • Developmental Outcomes: x_34 (Bayley Mental), x_35 (Bayley Motor), x_57 (Head Size
   1yr), y_factual (Child IQ Score 1yr).

## Guidelines
1.  Structure: Organize the note logically into smooth, narrative paragraphs (no
tables or bullet lists within sections):
 • Maternal History: Summarize demographics, habits, and medical history.
 • Paternal History: Briefly mention father's age and education.
 • Obstetric & Birth History: Detail past pregnancies, current delivery metrics, and
   neonatal status.
 • Infant Health Course: Describe feeding, lab values, and system-specific
   conditions.
 • Developmental Assessment: Report physical growth and developmental scores
   (critically y_factual Child IQ Score).
2.  Tone: Use formal medical terminology.
3.  Format: Write in continuous natural language.
4.  Data Integrity (CRITICAL): Transcribe numerical values EXACTLY as they appear.
DO NOT round or interpret qualitatively.
5.  Privacy: Refer to subjects as "the mother", "the infant", or "the patient".

## Input Data
The following attributes are recorded in the database:
> {features}

## Output
Write ONLY the body of the Medical Note. Do not include headers. Start directly
with the patient summary.
```

**Extraction prompts for** $(\tilde{X}, T, Y)$**.** We use GPT-4o-mini to parse each post into a JSON object containing $(\tilde{X}, T, Y)$, where $\tilde{X}$ is restricted to pre-treatment information and explicitly excludes outcome-revealing content to reduce leakage. We provide the dataset-specific prompt templates below (with consistent output schemas) to ensure comparable extraction across benchmarks.

**IHDP-post prompt: extract** $(\tilde{X}, T, Y)$

You are an expert in early childhood development who takes social media posts
written by mothers and converts them into structured information stored in a JSON
dictionary.

## Your Instructions
A user will provide a post and you must return a **valid JSON object** containing the
following keys along with the corresponding accurate information:
 • "x": Generate a single natural English sentence that includes only confirmed
   covariate information from the post (no guessing). Your goal is **HIGH RECALL**:
   capture as many explicitly stated covariates as possible. Extract covariates
   only when explicitly mentioned, using these names:
    -- x1: birthweight (kg); x2: head circumference (cm); x3: pre-term weeks; x4:
       birth order; x5: neo-natal health index; x6: mother's age; x7: child is
       female; x8: child is a twin; x9: mother married at birth; x10: mother left
       high school; x11: mother completed high school; x12: mother attended some
       college; x13: child is first born; x14: mother smoked; x15: mother consumed
       alcohol; x16: mother used drugs; x17: mother worked; x18: mother received
       prenatal care; x19--x25: living at site 1--7.
 • "treatment": Output "Yes" if participating in the early intervention program/IHDP,
   "No" if not, and "Unknown" if not mentioned.
 • "y_factual": Output the child's cognitive development score as a number if
   explicitly stated, otherwise "Unknown".

 **Critical Rules**:
1. Assign a valid value to each key above---never omit a key from the JSON.
2. Return **only the valid JSON object** (no extra text, explanations, or formatting).
3. For the "x" key: include ONLY covariates explicitly supported; pack as many facts
   as possible into **ONE** sentence using commas/semicolons.
4. Preserve numeric values exactly as written (including negatives and decimals).
5. Do not include treatment or outcomes inside "x".

## Extraction Guidance (for higher recall)
Treat common paraphrases as explicit: *married/single, graduated/left school,*
*smoked/cigarettes, boy/girl, first born, prenatal care/doctor visits.* If a post
lists ''did not'' items, include each as a separate clear fact.

## Examples
**Input**: *My baby boy arrived 1.13 weeks early, weighing −0.53 kg... We joined the*
*IHDP, and his score is 11.27.*
**Output**:
{ "x": "The child was born weighing −0.53 kg, the child was 1.13 weeks pre-term...
and the child is first born.", "treatment": "Yes", "y_factual": 11.27 }

**ACIC-post prompt: extract** $(\tilde{X}, T, Y)$

You are an expert in clinical data extraction who takes medical notes and converts
them into structured information stored in a JSON dictionary.

## Your Instructions
A user will provide a medical note and you must return a **valid JSON object**
containing the following keys along with the corresponding accurate information:
  • "x":  Generate a single natural English sentence that includes only confirmed
    covariate information from the note (no guessing).  Extract these covariates:
    x_1(Mother's age), x_2(Mother's marital status), x_3(Cigs/Day), x_4(Years Smoked),
    x_5(Height), x_6(Pre-preg Weight), x_7(Cardio Cond), x_8(Pulm Cond), x_9(Hema Cond),
    x_10(Endo Cond), x_11(Veneral Cond), x_12(Urin Cond), x_13(Gyne Cond), x_14(Neur
    Cond), x_15(Obst Compl), x_16(Infect Dis), x_17(Work Status), x_18(Years Educ),
    x_19(Income), x_20(Housing Density), x_21(Birth Place), x_22(Consanguinity),
    x_23(SES Index), x_24(Race), x_25(Age Menarche), x_26(Diastolic BP), x_27(Mom
    Birth Weight), x_28(Dad Age), x_29(Dad Years Educ), x_30(Prev Premes), x_31(Prev
    Abortions), x_32(Prev Pregs), x_33(Prev Stillbirths), x_34(Bayley Mental),
    x_35(Bayley Motor), x_36(Placental Weight), x_37(Cord Length), x_38(Infant
    Sex), x_39(Apgar 1m), x_40(Apgar 5m), x_41(Bottle Days), x_42(Breast Days),
    x_43(Bilirubin), x_44(Hematocrit), x_45(Hemoglobin), x_46(Neur Abn), x_47(CNS Cond),
    x_48(Muscoskel), x_49(Resp Abn), x_50(Cardio Abn), x_51(Liver Abn), x_52(Hema Cond),
    x_53(Infect Dis), x_54(Syndrome), x_55(Endo Dis), x_56(Medical Procs), x_57(Head
    Size 1yr), x_58(Gestational Age).
  • "treatment":  1 if High Birth Weight Category; 0 if Low Birth Weight Category.
  • "y_factual":  The outcome value (Child IQ Score at age 1) as a number.

**Critical Rules**:
1. Assign a valid value to each key|never omit a key from the JSON.
2. Return **only the valid JSON object** (no extra text or explanations).
3. For "x", strictly omit unmentioned covariates.

## Input
> The mother is a 30-year-old female, currently in a common-law relationship...  The
Child IQ Score at age 1 is recorded as 5.3080.

## Output
{ "x":  "The mother is 30 years old, in a common-law relationship...  and no other
reported abnormalities.", "treatment":  1, "y_factual":  5.3080 }

**Hillstrom prompt: extract** $(\tilde{X}, T, Y)$

You are an expert in online platforms who takes social media posts and converts them into structured information stored in a JSON dictionary.

## Your Instructions
A user will provide a post and you must return a **valid JSON object** containing the following keys along with the corresponding accurate information:
  • "x": Generate a single natural English sentence that includes only confirmed covariate information from the note (no guessing).  Extract these covariates: **recency** (months since last purchase), **history** (USD spent in the past year), **mens** (purchased men's merchandise), **womens** (purchased women's merchandise), **zip_code** (Urban/Suburban/Rural), **newbie** (new customer in the past year), **channel** (Phone/Web/Multichannel).  Omit any covariate not explicitly mentioned.  Never include visit status in this sentence.
  • "email_type":  "Yes" if received a marketing e-mail; "No" if explicitly stated not receiving them; "Unknown" if no mention.
  • "visit":  "Yes" if visited the website recently; "No" if explicitly stated not visiting; "Unknown" if no mention.

**Critical Rules**:
1. Assign a valid value to each key above|never omit a key from the JSON.
2. Return **only the valid JSON object** (no extra text, explanations, or formatting).
3. For the "x" key, strictly omit unmentioned covariates (no "Unknown" or inferences).

## Examples
**Input:**  *I'm new to this online shop!  Made my first purchase 2 months ago via phone, spent $120 on men's clothes.  Haven't gotten any marketing emails yet.*
**Output:**
{ "x":  "The user's last purchase was 2 months ago, they spent $120 in the past year, purchased men's merchandise, are a newbie customer, and used the Phone channel for shopping.", "email_type":  "No", "visit":  "Unknown" }

**RetailHero prompt: extract** $(\tilde{X}, T, Y)$

You are an expert Data Scientist specializing in Causal Inference and Uplift Modeling. Your task is to extract structured data from social media posts to build a ``User Profile'' (X), a ``Treatment'' (T), and an ``Outcome'' (Y).

## The ``Time Travel'' Rule (CRITICAL) You must strictly separate the **Pre-treatment History (X)** from the **Current Interaction (T & Y)**.
 • Imagine you are looking at the user's database record **ONE DAY BEFORE** this post was written.
 • The x field must describe the user **as they existed yesterday**, strictly ignoring what they did or received today.

## Your Instructions Return a **valid JSON object** with these keys:
"x": Generate a single natural English sentence summarizing the user's **historical profile** at time T−1.
 • **Recency:** Months since the *previous* purchase. (If they bought something today, ignore it and find the one *before*.)
 • **History:** Total USD spent in the past year. (Exclude today's spending.)
 • **Frequency:** Number of *past* visits. (``10th visit'' becomes ``9 previous visits''.)
 • **Demographics/Status:** Zip code type, gender preference, and newbie status.
**Forbidden Phrases in "x":**
 • NEVER say ``received an SMS'' or ``marketing text'' (This is T).
 • NEVER say ``just bought'' or ``made a purchase recently'' (This is Y=1).
 • NEVER say ``has not purchased recently'' (This is Y=0). State the specific historical time instead.
"sms_communication": ``Yes'' if the user explicitly mentioned receiving marketing communication via SMS in this interaction; otherwise ``No''.
"purchase": ``Yes'' if the user indicates they made *another* purchase during this specific interaction (Y=1); otherwise ``No''.

## Examples **Input:** *I just made my 10th visit! Spent $120 today. Shopping here for 2 months via phone. Got a text too.*
**Output:**
{ "x": "The user has made 9 previous visits, has been a customer for 2 months, purchases men's merchandise, and uses the Phone channel.", "sms_communication": "Yes", "purchase": "Yes" }

**Sema vs. Tirz prompt: extract** $(\tilde{X}, T, Y)$

You are a medical assistant supporting a physician who reviews Reddit posts
about weight loss treatments.  Translate each post into a structured JSON object
containing exactly three keys:  "x", "t", and "y".

## Output rules
 • "x":  Compose a single natural English sentence summarizing **ONLY** the user's
   pre-treatment covariates.  Include:  age, sex, country, t2dm (Type 2 Diabetes),
   metformin use, initial BMI, start_HbA1c, and start_weight.
   **CRITICAL:** Do **NOT** include any information about weight loss results, weight change,
   final weight, final HbA1c, side effects, or duration of treatment.  "x" must
   represent the state of the user **BEFORE** they took the medication.
 • "t":  The treatment the user reports, chosen from ["Tirzepatide", "Semaglutide",
   "Mounjaro", "Ozempic", "Wegovy", "Zepbound", "Rybelsus"].  If the treatment is not
   in this list, use "Unknown".
 • "y":  Set to "Yes" **ONLY** if the user reports achieving a weight loss of 5% or more
   (or provides numbers indicating they reached this threshold).  Otherwise, respond
   "No".

Return only the JSON dictionary|no other text.  Ensure "x" is a single sentence.

**Critical Rules:**
1. Strictly omit unmentioned covariates from "x" (do not use "Unknown" or guess).
2. "x" must be neutral and pre-treatment only.
3. If no pre-treatment covariates are found, set "x" to "No demographic details
   reported."

## Examples
**Input**
Subreddit:  r/Mounjaro
Comment:  SW 196, CW 192 (Female age 48 5'2") – Not diabetic.  Using for weight loss.
I felt a reduction in appetite but the loose stool was annoying.
**Output**
{"x":  "The user is a 48-year-old Female from the United States with a starting
weight of 196 who is not diabetic.", "t":  "Mounjaro", "y":  "No"}

**Input**
Subreddit:  r/Semaglutide
Comment:  57 year old female, started at 191 lbs.  Just took my 8th injection, down
14 lbs!  This med is a game changer.
**Output**
{"x":  "The user is a 57-year-old Female with a baseline weight of 191.", "t":
"Semaglutide", "y":  "Yes"}

**Input**
Subreddit:  r/Ozempic
Comment:  I am T2DM and taking Metformin.  Started Ozempic 0.5mg last week.  My A1C
was 8.2 start.  No weight loss yet.
**Output**
{"x":  "The user has T2DM, takes Metformin, and had a starting A1C of 8.2.", "t":
"Ozempic", "y":  "No"}

**Sema vs. Lira prompt: extract** $(\tilde{X}, T, Y)$

You are a medical assistant supporting a physician who reviews Reddit posts
about weight loss treatments. Translate each post into a structured JSON object
containing exactly three keys: "x", "t", and "y".

## Output rules
- "x": Compose a single natural English sentence summarizing **ONLY** the user's
  **PRE-TREATMENT** profile. Include these if mentioned: age, sex, country, t2dm (Type
  2 Diabetes), metformin use, initial BMI, start_HbA1c, and start_weight.
  **CRITICAL:** Do **NOT** include any information about weight loss results, weight change,
  final weight, final HbA1c, side effects, or duration of treatment. "x" must
  strictly represent the state of the user **BEFORE** they took the medication.
- "t": The treatment the user reports, chosen from ["Semaglutide", "Liraglutide",
  "Ozempic", "Wegovy", "Rybelsus", "Saxenda", "Victoza"]. If the treatment is not
  in this list or cannot be determined, use "Unknown".
- "y": Set to "Yes" **ONLY** if the user reports achieving a weight loss of **10% or more**
  of their starting weight (or provides numbers indicating they reached this 10%
  threshold). Otherwise, respond "No".

Return only the JSON dictionary---no other text. Ensure "x" is a single sentence.

**Critical Rules:**
1. Strictly omit unmentioned covariates from "x" (do not use ``Unknown'' or guess).
2. "x" must be neutral and pre-treatment only. Refer to the drug only as ``the
   treatment'' or ``the medication''.
3. If no pre-treatment covariates are found, set "x" to ``No baseline demographic or
   clinical details reported.''

## Examples
**Input**
Subreddit: r/liraglutide
Comment: My highest weight being higher than what I started Saxenda on at 174.5 lbs
and now down to 170.1 lbs. I also smoke weed and struggle with munchies. I'm on
day 4 of 1.8mg and feel sick.
**Output**
{"x": "The user has a starting weight of 174.5 lbs and reports a history of smoking
cannabis.", "t": "Saxenda", "y": "No"}

**Input**
Subreddit: r/Semaglutide
Comment: 57 year old female, 5'5''. Start weight 191 lbs. Just took my 8th
injection, down 20 lbs! This med is a game changer.
**Output**
{"x": "The user is a 57-year-old Female with a height of 5'5'' and a baseline weight
of 191 lbs.", "t": "Semaglutide", "y": "Yes"}

**Eren vs. Topi prompt: extract** $(\tilde{X}, T, Y)$

You are a medical assistant supporting a physician who reviews Reddit posts about
migraine treatments.  Translate each post into a structured JSON object containing
exactly three keys:  "x", "t", and "y".

## Output rules
 • "x":  Compose a single natural English sentence summarizing **ONLY** the user's
   **PRE-TREATMENT** profile.  Include these if mentioned:  age, sex, country, years of
   migraine suffering, migraine type (e.g., with aura, chronic, episodic), baseline
   MMD (days per month **BEFORE** this drug), and pre-existing conditions.
   **CRITICAL:** Do **NOT** include side effects of the current treatment, final MMD, or any
   ``after-treatment'' results.  "x" must strictly represent the state of the user
   **BEFORE** they took the drug in "t".
 • "t":  The treatment the user reports, chosen from ["Erenumab", "Topiramate",
   "Aimovig", "Topamax", "Epitomax", "Topiragen", "Eprontia", "Qudexy", "Trokendi"].
   If not in list, use "Unknown".
 • "y":  Set to "Yes" if the user experienced a **TREATMENT FAILURE** due to adverse
   effects (e.g., stopped, discontinued, intolerable side effects, switched drugs
   due to intolerance).  Otherwise, respond "No".

Return only the JSON dictionary---no other text.  Ensure "x" is a single sentence.

**Critical Rules:**
1. If no specific covariates are found, describe the user's migraine history.
2. Only if the post contains ZERO patient information, use ``No demographic or
   clinical details reported.''
3. Strictly omit unmentioned covariates from "x" (do not guess).

## Examples
**Input**
Comment:  I'm a 32F from the US. I've had chronic migraines with aura for 15 years.
Started Aimovig and it was a nightmare; the constipation was so bad I had to switch
to another med after a month.
**Output**
{"x":  "The user is a 32-year-old Female from the United States who has suffered from
chronic migraines with aura for 15 years.", "t":  "Aimovig", "y":  "Yes"}

**Input**
Comment:  Background:  45 male.  I've dealt with episodic migraines and anxiety
since my 20s.  I've been on Topamax 50mg for half a year now.  I feel a bit sleepy
but I'm staying on it because it works.
**Output**
{"x":  "The user is a 45-year-old Male with a history of episodic migraines and
anxiety since his 20s.", "t":  "Topamax", "y":  "No"}

**Onab vs. Topi prompt: extract** $(\tilde{X}, T, Y)$

```
You are a medical assistant aiding a physician.  Your role is to examine Reddit
posts discussing migraine treatments and convert self-reported information into a
structured JSON dictionary containing exactly three keys:  "x", "t", and "y".

## Your Instructions
A user will provide a Reddit post (including title and date).  You must return a
valid JSON object based on the following rules:
 • "x" (Baseline Covariates):  Generate a single natural English sentence summarizing
   the user's profile BEFORE they started the current treatment.
     -- Include:  age, sex, country, pregnancy status, pre-existing history (ADHD,
        anxiety, BP), and baseline MMD (monthly migraine days).
     -- CRITICAL PROHIBITION: Do NOT include side effects (tingling, fog), outcomes
        (''headaches decreased''), or medication names.
     -- Formatting:  Use integers for age and MMD; use floats for dosage (e.g.,
        50.0mg).
 • "t" (Treatment):  Identify the drug.  Allowed:  "Botox", "Topiramate", "Topamax",
   "Epitomax", "Topiragen", "Eprontia", "Qudexy", "Trokendi".  Otherwise, use
   "Unknown".
 • "y" (Cessation Status):  "Yes" if stopped specifically due to side effects; "No"
   if continued, stopped for other reasons, or not mentioned.

Critical Rules:
1. Return ONLY the valid JSON object.  No markdown, no explanations.
2. "x" must be a single string containing exactly one sentence.
3. Decoupling:  If I read only "x", I should have no idea which drug was taken or the
   outcome.

## Examples
Input:  I'm 26F from the US. I tried Topiramate before when I was having 17 migraine
days a month...  It made me really depressed so I stopped immediately.
Output:
{ "x":  "The user is a 26-year-old Female from the United States who had a baseline
of 17 migraine days per month.", "t":  "Topiramate", "y":  "Yes" }
```

# E. Additional Experimental Results

## E.1. Additional Seed-Stability Results

Figure 6a and Figure 6b provide additional raincloud plots on IHDP and Sema vs. Tirz, complementing the main-text results in Figure 3c and Figure 3d. Following the same protocol (fixed $k = 8$ and shared random seeds), SPIKED-CFR remains consistently more stable than PCA and RP, with lower median error and smaller dispersion across runs.

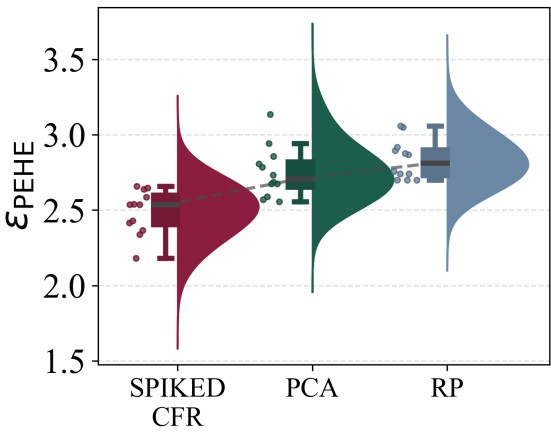
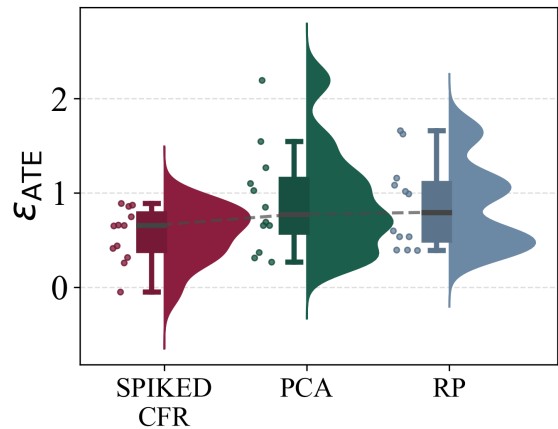



*(a) $\epsilon_{\text{PEHE}}$ on IHDP.*       *(b) $\epsilon_{\text{ATE}}$ on Sema vs. Tirz.*

*Figure 6.* Additional raincloud plots for seed stability on IHDP and Sema vs. Tirz.



### E.2. Projection dimension: absolute metrics

We report the absolute errors underlying Figure 3 for RP, PCA, and SPIKED-CFR across $k \in \{8, 16, 32, 64, 128\}$ (mean±std over random seeds); the main text uses relative gains since $\epsilon_{\text{PEHE}}$ and $\epsilon_{\text{ATE}}$ are not directly comparable.



*Table 5.* Absolute errors vs. $k$ for RP.

| Dataset | Metric | $k = 8$ | $k = 16$ | $k = 32$ | $k = 64$ | $k = 128$ |
|---|---|---|---|---|---|---|
| IHDP | $\epsilon_{\text{PEHE}}\downarrow$ | $2.78 \pm 0.18$ | $2.56 \pm 0.33$ | $2.56 \pm 0.33$ | $2.58 \pm 0.18$ | $2.98 \pm 0.18$ |
| ACIC | $\epsilon_{\text{PEHE}}\downarrow$ | $2.85 \pm 0.65$ | $2.42 \pm 0.39$ | $2.40 \pm 0.35$ | $2.42 \pm 0.19$ | $2.75 \pm 0.11$ |
| Sema vs. Tirz | $\epsilon_{\text{ATE}}\downarrow$ | $0.68 \pm 0.62$ | $1.22 \pm 1.10$ | $0.85 \pm 0.23$ | $0.59 \pm 0.63$ | $0.82 \pm 0.57$ |
| Eren vs. Topi | $\epsilon_{\text{ATE}}\downarrow$ | $1.53 \pm 0.95$ | $1.75 \pm 0.25$ | $3.45 \pm 1.32$ | $3.45 \pm 0.17$ | $3.83 \pm 1.93$ |





*Table 6.* Absolute errors vs. $k$ for PCA.

| Dataset | Metric | $k = 8$ | $k = 16$ | $k = 32$ | $k = 64$ | $k = 128$ |
|---|---|---|---|---|---|---|
| IHDP | $\epsilon_{\text{PEHE}}\downarrow$ | $2.79 \pm 0.19$ | $2.58 \pm 0.20$ | $2.58 \pm 0.19$ | $2.58 \pm 0.18$ | $2.74 \pm 0.19$ |
| ACIC | $\epsilon_{\text{PEHE}}\downarrow$ | $2.52 \pm 0.75$ | $2.53 \pm 0.44$ | $2.47 \pm 0.30$ | $2.62 \pm 0.89$ | $2.46 \pm 0.10$ |
| Sema vs. Tirz | $\epsilon_{\text{ATE}}\downarrow$ | $1.03 \pm 0.63$ | $1.67 \pm 0.57$ | $1.39 \pm 1.86$ | $0.62 \pm 0.52$ | $0.96 \pm 0.06$ |
| Eren vs. Topi | $\epsilon_{\text{ATE}}\downarrow$ | $1.83 \pm 1.25$ | $2.23 \pm 1.72$ | $3.72 \pm 2.58$ | $4.03 \pm 2.05$ | $3.94 \pm 1.17$ |





*Table 7.* Absolute errors vs. $k$ for SPIKED-CFR.

| Dataset | Metric | $k = 8$ | $k = 16$ | $k = 32$ | $k = 64$ | $k = 128$ |
|---|---|---|---|---|---|---|
| IHDP | $\epsilon_{\text{PEHE}}\downarrow$ | $2.48 \pm 0.17$ | $2.53 \pm 0.10$ | $2.48 \pm 0.19$ | $2.47 \pm 0.20$ | $2.50 \pm 0.14$ |
| ACIC | $\epsilon_{\text{PEHE}}\downarrow$ | $2.18 \pm 0.04$ | $2.33 \pm 0.16$ | $2.40 \pm 0.29$ | $2.34 \pm 0.24$ | $2.24 \pm 0.09$ |
| Sema vs. Tirz | $\epsilon_{\text{ATE}}\downarrow$ | $0.54 \pm 0.34$ | $1.14 \pm 0.91$ | $0.57 \pm 0.35$ | $0.51 \pm 0.26$ | $0.58 \pm 0.37$ |
| Eren vs. Topi | $\epsilon_{\text{ATE}}\downarrow$ | $1.27 \pm 0.72$ | $1.23 \pm 1.68$ | $3.14 \pm 1.90$ | $3.34 \pm 1.24$ | $3.74 \pm 1.74$ |



### E.3. Layer selection: unnormalized results

We provide the unnormalized metrics underlying Figure 4. Errors are reported as mean±std across random seeds, and the Wasserstein discrepancy is computed on the learned WPP projection.

*Table 8.* Layer-wise results on semi-synthetic benchmarks (unnormalized).

| Layer | IHDP $\epsilon_{\text{PEHE}}\downarrow$ | ACIC $\epsilon_{\text{PEHE}}\downarrow$ | IHDP Wass. | ACIC Wass. |
|---|---|---|---|---|
| L-1 | $2.51 \pm 0.22$ | $2.28 \pm 0.13$ | 0.56 | 0.35 |
| L-2 | $2.53 \pm 0.15$ | $2.23 \pm 0.08$ | 0.53 | 0.33 |
| L-4 | $2.47 \pm 0.20$ | $2.18 \pm 0.04$ | 0.40 | 0.60 |
| L-8 | $2.61 \pm 0.12$ | $2.25 \pm 0.09$ | 0.54 | 0.33 |
| L-12 | $2.51 \pm 0.23$ | $2.21 \pm 0.06$ | 0.55 | 0.38 |
| LTH | $2.49 \pm 0.21$ | $2.23 \pm 0.08$ | 0.46 | 0.18 |

*Table 9.* Layer-wise results on real-world clinical benchmarks (unnormalized).

| Layer | Sema vs. Tirz $\epsilon_{\text{ATE}}\downarrow$ | Eren vs. Topi $\epsilon_{\text{ATE}}\downarrow$ | Sema vs. Tirz Wass. | Eren vs. Topi Wass. |
|---|---|---|---|---|
| L-1 | $0.54 \pm 0.22$ | $1.54 \pm 2.12$ | 0.16 | 0.17 |
| L-2 | $0.67 \pm 2.40$ | $1.59 \pm 0.41$ | 0.16 | 0.10 |
| L-4 | $0.51 \pm 0.26$ | $1.27 \pm 0.72$ | 0.21 | 0.20 |
| L-8 | $0.60 \pm 0.16$ | $2.08 \pm 3.09$ | 0.15 | 0.19 |
| L-12 | $0.52 \pm 0.20$ | $1.63 \pm 1.27$ | 0.16 | 0.19 |
| LTH | $0.61 \pm 0.19$ | $1.84 \pm 0.71$ | 0.17 | 0.14 |

### E.4. Hyper-parameters Sensitivity to $k$ and $\alpha$.

We report a lightweight sensitivity study on four datasets (IHDP, ACIC, Sema vs. Tirz, and Eren vs. Topi) by sweeping the projection dimension $k \in \{8, 16, 32, 64, 128\}$ and the balancing weight $\alpha \in \{10^{-5}, 10^{-4}, 10^{-3}, 10^{-2}, 10^{-1}, 1, 10\}$. We evaluate (i) estimation error (Figure 7a, Figure 8a) and (ii) treated–control discrepancy measured by a projected Wasserstein distance. For the $k$ *sweep*, the Wasserstein distance in Figure 7b is computed in $k$ dimensions between the projected representations of the $T=1$ and $T=0$ groups (i.e., the projection-pursuit Wasserstein used by SPIKED-CFR). For the $\alpha$ *sweep*, the Wasserstein distance in Figure 8b is computed between the two treatment groups using the raw LLM representations (before the adapter), again via the same projected-Wasserstein routine.

**Effect of $k$.** Across datasets, performance is broadly stable over a wide range of $k$ (Figure 7a). A notable empirical pattern is that the best-performing $k$ often coincides with the largest projected Wasserstein discrepancy (Figure 7b), suggesting that the most suitable projection dimension is the one that most clearly exposes the treated–control shift.

**Effect of $\alpha$.** Sweeping $\alpha$ shows the expected trade-off: too small $\alpha$ under-balances, while overly large $\alpha$ can over-regularize (Figure 8a). Interestingly, the selected $\alpha$ correlates with the magnitude of treated–control discrepancy measured on raw LLM representations (Figure 8b): datasets with larger projected Wasserstein distances tend to prefer larger $\alpha$, yielding the near-diagonal trend in the scatter plot.

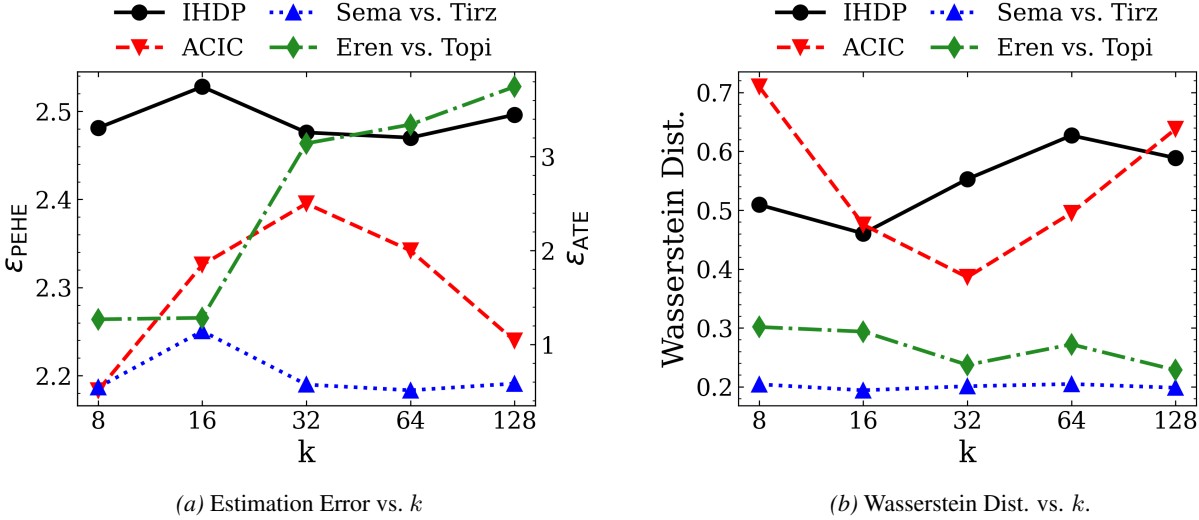

*(a)* Estimation Error vs. $k$        *(b)* Wasserstein Dist. vs. $k$.

*Figure 7.* $k$ **sensitivity.** $k$ values that induce larger projected Wasserstein distances typically yield lower $\epsilon_{\text{PEHE}}$ and $\epsilon_{\text{ATE}}$.

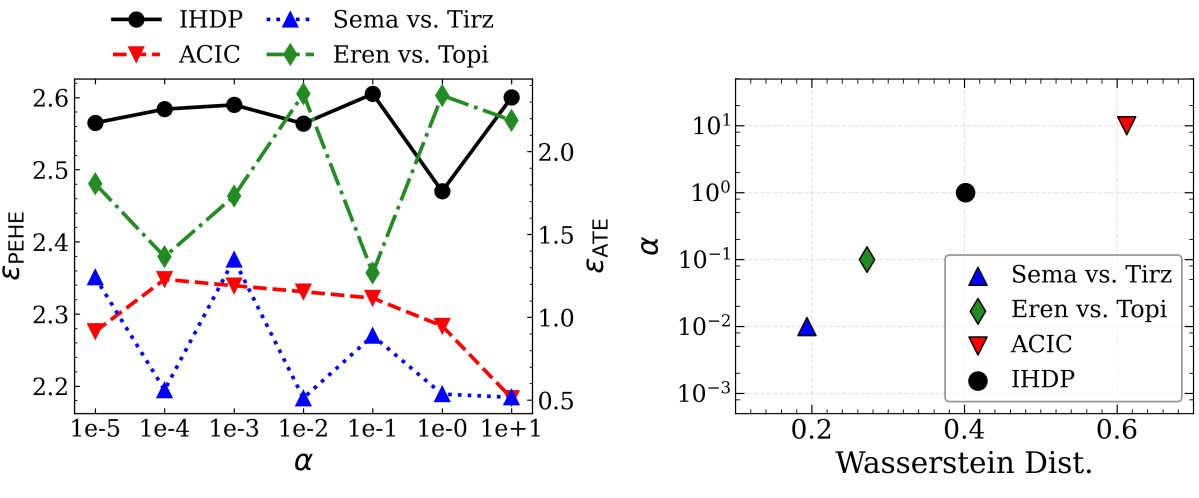

*(a)* Estimation Error vs. $\alpha$        *(b)* Correlation between $\alpha$ and Wasserstein Dist.

*Figure 8.* $\alpha$ **sensitivity.** Preferred $\alpha$ increases with treated–control discrepancy measured by projected Wasserstein distance.

