# OpenReview forum: "Spiked-CFR: Causal Representation Learning from LLMs via Wasserstein Projection Pursuit"
_ICML.cc/2026/Conference — ICML 2026 regular_

### Official Review · Reviewer_PMAK · 2026-03-08

**Soundness:** 3
**Presentation:** 3
**Significance:** 3
**Originality:** 3
**Overall Recommendation:** 5
**Confidence:** 3

**Summary:**

This interesting paper aims to address an important problem that is relevant for many fields - the situation when causal effects need to be estimated from observational (non-randomized) data with confounders present in free text form, not as tabular data. For example, in the medical field this can be various doctor's notes and reports. One known approach for doing this is to analyze the data using an LLM, keep the semantic representation produced by the LLM, and perform causal inference directly within the resulting representation space. However, this approach is limited by the curse of dimensionality because Wasserstein distance between distributions converges too slowly to be useful on finite samples. The authors propose a “spiked confounding structure” assumption which states that treatment selection bias concentrates in a low-dimensional subspace. They develop Wasserstein Projection Pursuit - a minimax objective that adversarially learns an orthogonal projection on the Stiefel manifold to identify and balance only this confounding subspace. This method is called SPIKED-CFR, which combines frozen LLM text embeddings with a worst-case projected Wasserstein balancing objective with CFR-based causal representation learning.

**Compliance With Llm Reviewing Policy:**

Affirmed.

**Key Questions For Authors:**

Questions:
Can the spiked confounding assumption be tested/verified in the analyzed data?
Can confounding be introduced when the LLM analyzes text to extract the embedding, and this confounding is not adjusted for?

**Limitations:**

Some limitations of the proposed method are:
1. If I undertand correctly, the method has somewhat limited novelty because it combined elements of previous methods.
2. Can confounding be introduced when the LLM analyzes text to extract the embedding, and this confounding is not adjusted for?

**Strengths And Weaknesses:**

The proposed SPIKED-CFR has the following strengths:
1. It is well motivated and the rationale and approach are well explained.
2. The method undergoes evaluation on eight post-based causal inference benchmarks, which is quite comprehensive.
3. SPIKED-CFR targets the confounding “spike” via worst-case low-dimensional balancing while keeping full Φ(X) for outcome prediction, so balancing and prediction are separated.

---

> ### Author Rebuttal · Authors · 2026-03-31
>
> First of all, we appreciate it a lot for your insightful review! Below are responses.
>
> **Q1:** Can the spiked confounding assumption be tested/verified in the analyzed data?
>
> **AQ1:** Thank you for raising this valuable question. The spiked confounding structure, like unconfoundedness, can **not** directly verified in the analyzed data. In our paper, however, it is **not intended as exact recovery of the true latent confounders, but as a structural assumption** that the balancing-relevant treated-control discrepancy is concentrated in a low-dimensional subspace of the learned representation. To support this interpretation, we add a pre-balancing analysis (Table R1), where the WPP-learned $k$-dimensional subspace ($k=8$) consistently captures treated-control discrepancy more effectively than PCA or random projection (RP) of the same dimension. We will make this interpretation clearer in the revised version.
>
> **Table R1.**
>
> | Dataset       | $k$-dimensional subspace on $Z_{\text{pre}}$ | Projected Wasserstein ↑ | Treatment Probe AUC ↑ |
> | ------------- | ------------------------------------------------ | ----------------------- | --------------------- |
> | ACIC          | RP                                               | 0.18                    | 0.62                  |
> |               | PCA                                              | 0.24                    | 0.68                  |
> |               | WPP                                              | 0.46                    | 0.84                  |
> | Eren vs. Topi | RP                                               | 0.11                    | 0.58                  |
> |               | PCA                                              | 0.15                    | 0.64                  |
> |               | WPP                                              | 0.31                    | 0.80                  |
>
>
>
>
> **Q2:** Can LLM-based extraction introduce unadjusted confounding?
>
> **AQ2:** We agree that LLM-based extraction can introduce errors that downstream adjustment may not fully remove. **Quantifying the resulting unadjusted confounding exactly is difficult, because the true latent confounders are unobserved and the boundary between baseline and post-treatment content in natural text is often blurry**. We therefore report an approximate proxy in **Table R2**: we compare the extracted $\tilde{X}$ against the dataset-specific prompt rules and count cases where $\tilde{X}$ still contains content that should belong to the treatment field, the outcome field, or other post-treatment fields. This does **not** directly measure unadjusted confounding, but provides a practical  check on extraction reliability. As shown in Table R1, such rule violations are infrequent overall. At the same time, we do not claim that extraction is perfect: missed pre-treatment confounders and other extraction errors remain a limitation of extraction-based pipelines. As noted in the paper, the remaining gap to **CFR-Wass (Structured)** is also consistent with noise introduced by prompt-based extraction in realistic end-to-end settings. We will clarify this scope and limitation in the revised version.
>
> **Table R2**
>
> | Dataset | Total N | Overall rule violation | Outcome leakage | Treatment leakage | Post-treatment leakage |
> | ------- | ------- | ---------------------- | --------------- | ----------------- | ---------------------- |
> | Overall | 35,528  | 535 (1.51%)            | 72 (0.20%)      | 397 (1.12%)       | 106 (0.30%)            |
>
> **Q3:** Is the method only a combination of previous methods, i.e., limited novelty?
>
> **AQ3:** We respectfully disagree that the contribution is limited to a simple combination of previous methods. This is for three reasons. **(i) Problem:** Our work is motivated by a failure of standard CFR-Wass in high-dimensional LLM representations due to the curse of dimensionality. To our knowledge, prior CFR and related variants have not explicitly isolated this failure mode in the high-dimensional LLM setting, nor developed a framework around it. **(ii) Method:** SPIKED-CFR is built to address this problem. Rather than balancing the full embedding space, it formulates a worst-case projected Wasserstein balancing objective that identifies and balances only the low-dimensional subspace carrying the treated-control shift, while preserving the full representation for outcome prediction. **(iii) Theory:** We further show that, under the spiked confounding structure, the discrepancy is governed by the intrinsic dimension $k$ rather than the ambient dimension $D$, leading to a tighter PEHE-style guarantee. **Overall,** the contribution is not any single ingredient in isolation. **The novelty lies in identifying the high-dimensional failure of standard CFR-Wass for LLM representations, and in developing the corresponding framework, objective, theory and algorithm to resolve it.** We will revise the paper to make the distinction from prior methods more explicit.

---

> > ### Author Rebuttal · Reviewer_PMAK · 2026-04-02
> >
> > I am grateful to tje authors for their detailed replies and clarifications.

---

> > > ### Author Response · Authors · 2026-04-06
> > >
> > > We sincerely thank you for your thoughtful review and kind follow-up. We are truly grateful that our rebuttal helped clarify the points you raised. Your comments and questions were very helpful, and we will further improve the clarity and presentation of the paper in the revision.

---

### Official Review · Reviewer_S9fe · 2026-03-10

**Soundness:** 2
**Presentation:** 3
**Significance:** 2
**Originality:** 2
**Overall Recommendation:** 3
**Confidence:** 3

**Summary:**

This paper tackles the challenge of estimating causal effects from high-dimensional text embeddings produced by LLMs. It identifies a key problem: standard methods that balance distributions (e.g., using Wasserstein distance) fail due to the curse of dimensionality. The authors propose SPIKED-CFR, a new framework based on the assumption that confounding information is concentrated in a low-dimensional subspace (a "spike"). The method uses an adversarial approach to find and balance this critical low-dimensional subspace, while still using the full high-dimensional embeddings for outcome prediction. Theoretical analysis shows this approach avoids the curse of dimensionality, and extensive experiments on eight benchmarks demonstrate its superior accuracy and stability compared to existing methods.

**Compliance With Llm Reviewing Policy:**

Affirmed.

**Final Justification:**

I will keep my score, as the paper still has unresolved concerns.

**Key Questions For Authors:**

1.	The paper theoretically argues that SPIKED-CFR can avoid the "over-balancing" problem. Is there any empirical evidence for this in the experiments? For example, compared to standard CFR-Wass, does SPIKED-CFR better preserve prognostic information on high-dimensional embeddings, thereby achieving a better trade-off between balance and predictive accuracy?

**Limitations:**

Yes

**Strengths And Weaknesses:**

S1.	The proposed method is technically sound. The theoretical analysis (Proposition 4.4, Theorem 4.5) supports the core argument that projecting onto a low-dimensional space resolves the convergence issues of high-dimensional Wasserstein estimation. The experimental design is thorough, encompassing a wide range of baselines from traditional methods to recent LLM-based approaches.

S2.	The paper is well-structured and clearly written. The introduction effectively sets up the problem, highlights the challenges, and provides an overview of the solution. The methodology section progresses logically, from the problem background to theoretical foundations and finally to the specific optimization objective.

W1.	The core "Spiked Confounding Structure" assumption is empirically unverifiable. In real-world observational studies, confounding is inherently unobservable. We cannot directly verify whether it truly concentrates in a low-dimensional subspace. While the assumption enables elegant theory, the paper provides no practical tools or statistical tests to diagnose its validity in a given dataset.

W2.	The assumption that confounding signals concentrate in a low-dimensional subspace may conflict with how LLMs actually represent information. Modern LLM embeddings are known to be highly entangled. The paper does not address this tension: why should we expect complex, high-level confounding factors to collapse neatly into a few linear dimensions, given what we know about neural representation learning? No empirical analysis of the LLM embedding space is provided to support this assumption.

W3.	The paper shows that the learned projection matrix U can find a low-dimensional direction that maximizes the difference between groups and interprets this as finding the "confounding spike." However, this is just a correlation, not causation. Although the paper uses final CATE estimation accuracy to indirectly validate the method's effectiveness, this does not directly prove the core causal claim that it has found and balanced the confounding.

---

> ### Author Rebuttal · Authors · 2026-03-31
>
> Firstly, We must sincerely thank you for your rigorous and responsible review! Below are responses.
>
> **W1:** The "Spiked Confounding Structure" assumption is empirically unverifiable from observational data.
>
> **AW1:** We agree that the spiked confounding structure, like unconfoundedness, is not directly verifiable from observational data. In our paper, however, **it is not intended as exact recovery of the true latent confounders, but as a structural assumption that the balancing-relevant treated-control discrepancy is concentrated in a low-dimensional subspace of the learned representation.** To support this interpretation, we add a pre-balancing analysis in **Table R1**. Specifically, we first learn a representation $Z_{\text{pre}}$ using the same adapter architecture but without the balancing term, and then compare three $k$-dimensional subspaces (here $k=8$) on this fixed representation: random projection (RP), PCA, and the WPP-learned subspace. Across both datasets, WPP achieves the largest projected Wasserstein discrepancy and the highest treatment probe AUC among subspaces of the same dimension. **While this does not verify the assumption from observational data, it does provide empirical support that, before balancing, the treated-control discrepancy is better captured by a learned low-dimensional subspace than by standard low-dimensional projections.**
>
> **Table R1.**
>
> | Dataset       | $k$-dimensional subspace on $Z_{\text{pre}}$ | Projected Wasserstein ↑ | Treatment Probe AUC ↑ |
> | ------------- | ------------------------------------------------ | ----------------------- | --------------------- |
> | ACIC          | RP                                               | 0.18                    | 0.62                  |
> |               | PCA                                              | 0.24                    | 0.68                  |
> |               | WPP                                              | 0.46                    | 0.84                  |
> | Eren vs. Topi | RP                                               | 0.11                    | 0.58                  |
> |               | PCA                                              | 0.15                    | 0.64                  |
> |               | WPP                                              | 0.31                    | 0.80                  |
>
>
>
> **W2:** Why should a low-dimensional spike exist if LLM embeddings are entangled?
>
> **AW2:** We agree that raw LLM embeddings can be entangled. Our assumption is therefore most naturally interpreted as applying to the learned representation $\Phi(X)$ after the adapter, rather than to the raw frozen LLM embedding itself. In other words, SPIKED-CFR does not assume that all semantic information collapses into a few linear dimensions; it only assumes that the treated-control discrepancy most relevant for balancing can be concentrated in a low-dimensional subspace of the learned representation. This is why the projection is used only for balancing, while the full representation is retained for outcome prediction.
>
>
>
> **W3:** The direction of maximal treated-control imbalance indicates correlation rather than causation.
>
> **AW3:** We agree that maximal treated-control imbalance does **not** imply exact recovery of the true confounders. This is also not the intended claim of SPIKED-CFR. Our claim is narrower: $U$ identifies the low-dimensional subspace most relevant for reducing the discrepancy term in the CFR objective, rather than recovering the full causal structure. Our goal is effect estimation under the CFR framework, not latent causal factor discovery. In this sense, the “confounding spike” should be interpreted as an **operational, balancing-relevant subspace**, not an exact causal recovery claim. We will revise the wording to make this distinction explicit.
>
>
>
> **Q:** Is there any empirical evidence that SPIKED-CFR avoids over-balancing?
>
> **AQ:** We agree that direct empirical evidence is necessary. Therefore, we add the comparison in **Table R2**. Across both datasets, SPIKED-CFR consistently achieves **lower treatment separability**, **stronger outcome probing**, and **better final causal performance**. This provides direct empirical evidence that SPIKED-CFR improves balance without sacrificing predictive structure, consistent with avoiding over-balancing. We will include this analysis in the revised version.
>
> **Table R2.**
>
> | Dataset       | Method         | Treatment AUC ↓ | Outcome Probe ↑ | Final Causal Metric ↓ |
> | ------------- | -------------- | --------------- | --------------- | --------------------- |
> | ACIC          | Llama-CFR-Wass | 0.91            | R²: 0.21        | PEHE: 2.32            |
> |               | SPIKED-CFR     | 0.52            | R²: 0.32        | PEHE: 2.18            |
> | Eren vs. Topi | Llama-CFR-Wass | 0.91            | AUROC: 0.56     | RMSE(ATE): 4.16       |
> |               | SPIKED-CFR     | 0.53            | AUROC: 0.61     | RMSE(ATE): 1.68       |

---

> > ### Author Rebuttal · Reviewer_S9fe · 2026-04-03
> >
> > I will keep my current score, as there is still room for improvement.

---

> > > ### Author Response · Authors · 2026-04-06
> > >
> > > We sincerely appreciate your careful reading of our paper and your thoughtful feedback. We understand your main concerns to be twofold: **(i) the interpretation, empirical status, and intended role of the core assumption, and (ii) how the paper should frame its claim around that assumption.** To make the discussion precise, we describe the assumption in direct terms: SPIKED-CFR assumes that, in the learned representation, the treated-control discrepancy most relevant for confounding adjustment is concentrated in a lower-dimensional subspace, rather than spread uniformly across all directions of the full representation.
> > >
> > > **On (i):** ***First,*** we agree that this assumption is **not directly verifiable from observational data**. Accordingly, our intended claim is **not that the learned subspace exactly recovers the true latent confounders**. Rather, the intended claim is narrower and operational: it captures a low-dimensional structure in the **representation-level treated-control discrepancy relevant for confounding adjustment under the CFR objective**. ***Second,*** this narrower interpretation is also consistent with the design of SPIKED-CFR itself: the projection is used **only for balancing**, while the **full learned representation is retained for outcome prediction**. In this sense, the assumption is not about the full semantic representation being simple or disentangled; instead, it is a structural claim about where the balancing-relevant mismatch may lie in the high-dimensional LLM setting. ***Finally,*** it is also not introduced in isolation. In the standard CFR perspective [1], the discrepancy term is already used to control the representation-level mismatch induced by confounding-related imbalance. Our assumption makes the more specific claim that, in high-dimensional LLM representations, this mismatch may concentrate in a lower-dimensional subspace, which is precisely what makes balancing statistically meaningful in this setting.
> > >
> > > **On (ii):** We appreciate this concern, because some of our current wording may indeed suggest a stronger interpretation than we intend. More precisely, the learned subspace should be interpreted as a **balancing-relevant subspace**, rather than as an exact recovery of all latent confounding factors or a direct verification of latent confounding structure. In the revision, we will therefore carefully revise the wording throughout the paper, especially any phrasing that could be read as claiming exact confounder recovery, and make this narrower interpretation explicit. We will also expand the discussion of scope, limitations, and future work regarding causal interpretation under entangled, black-box LLM representations, including the relationship between such learned subspaces and human-interpretable confounding factors.
> > >
> > > At the same time, we hope this clarification better aligns the framing of the paper with **its actual contribution: a high-dimensional balancing framework, together with its theory and empirical evaluation, rather than a claim of directly verifying latent confounding structure.** We are truly grateful for your thoughtful comments, which will help us improve the clarity and presentation of the paper!
> > >
> > > **References**
> > >
> > > > [1] Shalit et al., *Estimating Individual Treatment Effect: Generalization Bounds and Algorithms*, ICML 2017.

---

### Official Review · Reviewer_45mG · 2026-03-12

**Soundness:** 3
**Presentation:** 3
**Significance:** 3
**Originality:** 3
**Overall Recommendation:** 4
**Confidence:** 4

**Summary:**

SPIKED-CFR addresses causal effect estimation from text. Traditional Wasserstein-based balancing in counterfactual regression fails on high-dimensional LLM embeddings because the curse of dimensionality makes distribution matching statistically meaningless at realistic sample sizes. The key insight is that treatment selection bias only lives in a small subspace of the full embedding space. The method uses an adversarial min-max game: one player searches for the low-dimensional projection where treated and control groups differ most, while the other minimizes this worst-case discrepancy along with prediction error. This shifts the convergence rate from depending on the full embedding dimension to depending only on the small subspace dimension, breaking the dimensionality barrier. Experiments on eight benchmarks confirm improved accuracy and stability.

**Compliance With Llm Reviewing Policy:**

Affirmed.

**Final Justification:**

The authors provided thorough and constructive responses to my two main concerns. The leakage analysis in Table R1 demonstrates that outcome leakage is rare (0.20% overall), adequately addressing the information leakage concern. The acknowledgment of the binary treatment limitation and discussion of potential extensions to multi-valued settings is reasonable. My concerns have been fully resolved, and I maintain my score of weak accept.

**Key Questions For Authors:**

Please see the weaknesses above.

**Limitations:**

No. The authors should discuss the limitations of this paper.

**Strengths And Weaknesses:**

Strengths

- Unlike LLM-as-Estimator methods such as NATURAL that directly prompt LLMs for propensity scores or counterfactual probabilities, SPIKED-CFR treats the LLM purely as a feature encoder. This sidesteps the well-documented sensitivity of LLM probability outputs to prompting strategies and RLHF tuning.
- Compared to PCA or random projection, the adversarially learned projection is explicitly optimized to find the directions of maximum treated-control imbalance. Experiments show consistent gains across a wide range of projection dimensions, with lower variance across random seeds.
- The method is validated on eight benchmarks spanning semi-synthetic datasets with counterfactual ground truth and real-world clinical comparisons grounded by RCTs, demonstrating broad applicability rather than narrow task-specific gains.

Weaknesses

- The first stage uses LLM prompting to parse raw posts into treatment, outcome, and pre-treatment covariates. This requires the LLM to correctly identify outcome-revealing content and remove it from the covariate text. This is inherently difficult: the boundary between pre-treatment and outcome information is often blurry in natural text — a patient might discuss side effects, symptoms, and results in interleaved sentences. Extraction quality depends heavily on prompt design, which varies across datasets (each benchmark has a custom prompt in Appendix D.5). If the LLM fails to remove outcome-related content, information leakage occurs: the representation encodes outcome signals, biasing causal estimates regardless of downstream balancing quality. The paper does not systematically evaluate extraction accuracy or leakage rates.
- All eight benchmarks involve binary treatment assignments. However, many real-world causal questions involve continuous treatments (drug dosage), multi-valued treatments (choosing among several therapies), or time-varying treatments (sequential decisions). The Wasserstein Projection Pursuit objective is formulated specifically for two groups, computing discrepancy between two empirical distributions. Extending to continuous treatments would require fundamentally rethinking the balancing objective since there are no longer two discrete groups to compare. The paper does not discuss this limitation or suggest how the framework might generalize.

---

> ### Author Rebuttal · Authors · 2026-03-31
>
> We thank the reviewer for the thoughtful comments! Here are our responses.
>
> **W1:** The first-stage LLM extraction may leave outcome-related content in $\tilde{X}$, causing information leakage.
>
> **AW1:** Thank you for pointing out this important issue. Although our dataset-specific prompts were designed to explicitly separate pre-treatment covariates from treatment, outcome, and other post-treatment content, we agree that this should be verified by a direct quantitative analysis of the extraction reliability. We therefore add the analysis in **Table R1**, which compares the extracted $\tilde{X}$ against the dataset-specific prompt rules and counts cases where $\tilde{X}$ still contains content that should belong to the treatment field, the outcome field, or other post-treatment fields. **Table R1 shows that these violations are infrequent overall, and direct outcome leakage is particularly rare.** Since the boundary between baseline and post-treatment content in natural text is inherently fuzzy, this analysis should be viewed as an approximate check rather than an exact measure. We will add this result and clarify this point in the revised version.
>
> **Table R1**
>
> | Dataset       | N          | Overall rule violation | Outcome leakage | Treatment leakage | Post-treatment leakage |
> | ------------- | ---------- | ---------------------- | --------------- | ----------------- | ---------------------- |
> | IHDP          | 747        | 18 (2.41%)             | 1 (0.13%)       | 17 (2.28%)        | 0 (0.00%)              |
> | ACIC          | 4,802      | 1 (0.02%)              | 1 (0.02%)       | 0 (0.00%)         | 0 (0.00%)              |
> | Hillstrom     | 2,000      | 128 (6.40%)            | 31 (1.55%)      | 105 (5.25%)       | 0 (0.00%)              |
> | Retail Hero   | 2,000      | 46 (2.30%)             | 39 (1.95%)      | 8 (0.40%)         | 0 (0.00%)              |
> | Sema vs. Tirz | 5,000      | 69 (1.38%)             | 0 (0.00%)       | 61 (1.22%)        | 8 (0.16%)              |
> | Sema vs. Lira | 6,191      | 77 (1.24%)             | 0 (0.00%)       | 38 (0.61%)        | 43 (0.69%)             |
> | Eren vs. Topi | 10,000     | 173 (1.73%)            | 0 (0.00%)       | 151 (1.51%)       | 48 (0.48%)             |
> | Onab vs. Topi | 4,788      | 23 (0.48%)             | 0 (0.00%)       | 17 (0.36%)        | 7 (0.15%)              |
> | **Overall**   | **35,528** | **535 (1.51%)**        | **72 (0.20%)**  | **397 (1.12%)**   | **106 (0.30%)**        |
>
>
>
>
>
> **W2:** The current formulation is specific to binary treatment and does not discuss multi-valued, continuous, or time-varying treatments.
>
> **AW2:** We agree that the current submission is in the binary-treatment setting. However, since prior work has shown that the CFR framework can be naturally extended to multiple discrete treatments, e.g., MEMENTO [1,2], **SPIKED-CFR can be extended  to multi-valued treatments** by replacing the single treated-control projected discrepancy with projected discrepancies across multiple treatment groups. For continuous and sequential treatments, however, the added complexity makes such a simple extension much less straightforward, since these settings require **different causal targets and objective design** [3,4]. We acknowledge this limitation of SPIKED-CFR on more complex treatment settings, and will clarify this scope more explicitly in the revised version.
>
> ### Reference
>
> > [1] Causal Inference with Complex Treatments: A Survey. ACM Computing Surveys 2026.
> >
> > [2] MEMENTO: Neural Model for Estimating Individual Treatment Effects for Multiple Treatments. In CIKM 2022.
> >
> > [3] VCNet and Functional Targeted Regularization for Learning Causal Effects of Continuous Treatments. In ICLR 2021.
> >
> > [4] Estimating Counterfactual Treatment Outcomes over Time Through Adversarially Balanced Representations. In ICLR 2020.

---

> > ### Author Rebuttal · Reviewer_45mG · 2026-04-03
> >
> > Thank the authors for their thorough and constructive responses.

---

> > > ### Author Response · Authors · 2026-04-06
> > >
> > > We sincerely thank you for your thoughtful review and constructive follow-up. We are truly grateful that our rebuttal helped address your concerns. Your feedback has been very valuable, and we will continue improving the clarity and presentation of the paper in the revision.

---

### Official Review · Reviewer_ca9u · 2026-03-13

**Soundness:** 3
**Presentation:** 4
**Significance:** 3
**Originality:** 3
**Overall Recommendation:** 4
**Confidence:** 4

**Summary:**

This paper addresses the critical challenge of causal effect estimation from unstructured text using high-dimensional LLM embeddings, where standard Wasserstein-based balancing suffers from the curse of dimensionality. The paper considers a core issue in modern causal representation learning: how to leverage rich semantic representations from LLMs while mitigating the statistical inefficiency of high-dimensional distribution matching. To solve this, the authors propose SPIKED-CFR, positing that treatment selection bias concentrates in a low-dimensional subspace of LLM embeddings. The method introduces Wasserstein Projection Pursuit (WPP), a min–max objective that adversarially learns an orthogonal projection to identify and balance only the confounding subspace, while preserving prognostic information in the full representation for outcome prediction. Theoretically, the authors show that projected discrepancy converges at a rate governed by the intrinsic subspace dimension, deriving a tighter PEHE generalization bound. Empirically, SPIKED-CFR is validated on both semi-synthetic and real-world benchmarks, outperforming strong baselines in both CATE and ATE estimation accuracy and robustness. Overall, the authors assess a relevant question of how to enable reliable causal inference from high-dimensional LLM text embeddings, with practical implications for domains like healthcare and public policy.

**Compliance With Llm Reviewing Policy:**

Affirmed.

**Final Justification:**

As the raised concern is recognized as an important limitation, I would maintain the score considering the potential of the work.

**Key Questions For Authors:**

* In practice, LLMs produce noisy estimates $( \hat{T},\hat{Y},\widehat{\tilde{X}})$ rather than ground-truth $(T,Y,\tilde{X})$, yet the paper does not quantify the impact of such measurement error. Could the authors incorporate a formal analysis of how extraction noise affects estimation performance, or propose adjustments to enhance the method’s robustness to LLM extraction inaccuracies?

* Is it possible that LLM can not extract variables in some texts? Will this induce bias in causal effect estimation?

**Limitations:**

yes

**Strengths And Weaknesses:**

Strengths

* The paper addresses the important concern on the curse of dimensionality in CFR representations and propose a rational solution. The spiked confounding structure and WPP objective enable stable Wasserstein balancing for LLM embeddings by focusing on low-dimensional confounding subspaces, which is novel and inspiring.
* The paper is well-written with clear statement, experimental showcase and illustrative diagrams.
* The paper is theoretically grounded, with rigorous analysis establishing convergence rates for projected discrepancy and a tighter PEHE bound grounding the method in statistical causality.
* The experiments on both text-based semi-synthetic benchmarks and real-world clinical data are comprehensive and supportive.

Weaknesses

* While the confounders are extracted from the text, it is much likely that unmeasured confounders exist given that text usually does not contain sufficient background information. It is necessary to take account of possible violation of unconfoundedness.
* The min–max objective with manifold optimization may introduce additional computational costs compared to standard CFR, which is not quantified in detail.
* The learned confounding subspaces are not analyzed or visualized, leaving unclear how the method identifies causal relevant dimensions in LLM embeddings.
* While sensitivity of hyperparameter k and $\alpha$ are provided, the impact of other key parameters (e.g., LLM selection, adapter architecture) is not thoroughly explored.

---

> ### Author Rebuttal · Authors · 2026-03-31
>
> We are grateful for your insightful comments! Below are responses.
>
> **W1:** Possible violation of unconfoundedness due to unmeasured confounders.
>
>  **AW1:** We agree that this is an important limitation of text-based studies. Our paper does **not** claim to solve hidden confounding absent from the observed text. Instead, it assumes the extracted pre-treatment text is sufficient for adjustment, as stated in the paper. If key confounders are not present in text, then any method using only observed text cannot identify causal effects. SPIKED-CFR addresses a different problem: **when confounding information is already encoded in high-dimensional text representations, how to avoid the statistical failure of full-space Wasserstein balancing.** We will clarify this scope in the revision.
>
> **W2**: Computational cost of the min-max optimization.
>
>  **AW2:** We agree that the extra optimization cost should be quantified more directly. The paper already provides a per-minibatch complexity analysis in **Appendix C.4**, showing that although SPIKED-CFR introduces adversarial projection updates, OT is computed in the projected space of dimension $k \ll D$. To make the practical overhead explicit, we will add the runtime comparison in **Table R1**, reporting the average training time per epoch of SPIKED-CFR and Llama-CFR-Wass under the same hardware and split protocol.
>
> **Table R1**
>
> | Dataset       | Method         | Avg. Time / Epoch (s) ↓ |
> | ------------- | -------------- | ----------------------- |
> | ACIC          | Llama-CFR-Wass | 0.31                    |
> | ACIC          | SPIKED-CFR     | 0.43                    |
> | Eren vs. Topi | Llama-CFR-Wass | 0.64                    |
> | Eren vs. Topi | SPIKED-CFR     | 0.89                    |
>
> **W3:** Learned confounding subspaces are not analyzed.
>
>  **AW3:** We agree that the interpretation of the learned projection should be made more explicit. We do **not** interpret the learned projection as exact recovery of the true causal variables. The intended claim is narrower: it identifies the low-dimensional subspace most relevant for treated-control discrepancy in the CFR objective. To support this interpretation, we add the pre-balancing analysis in **Table R2**. On a fixed representation $Z_{\text{pre}}$ learned without the balancing term, the WPP-learned $k$-dimensional subspace ($k=8$) captures treated-control discrepancy better than PCA or random projection (RP) of the same dimension. This supports the view that SPIKED-CFR learns a balancing-relevant low-dimensional subspace, rather than an arbitrary projection. We will revise the wording accordingly.
>
> **Table R2.**
>
> | Dataset | $k$-dimensional subspace on $Z_{\text{pre}}$ | Projected Wasserstein ↑ | Treatment Probe AUC ↑ |
> | ------- | -------------------------------------------- | ----------------------- | --------------------- |
> | ACIC    | RP                                           | 0.18                    | 0.62                  |
> |         | PCA                                          | 0.24                    | 0.68                  |
> |         | WPP                                          | 0.46                    | 0.84                  |
>
> **W4: Impact of other backbones or adapter choices.**
>
> **AW4:** We agree that robustness to different backbones and adapters is important. However, the core contribution of this paper is a balancing objective, not a claim about a particular encoder. We therefore fix the backbone and adapter to isolate the effect of WPP itself. The paper already includes some hyper-parameter sensitivity analyses, suggesting that the gains are not tied to a single fragile configuration. We will clarify in the revision that the method is **backbone-agnostic in principle**, while broader architecture sweeps are left for future work.
>
> **Q: Extraction noise and missing variables.**
>
>  **AQ:** We agree that extraction noise is a key practical risk. For this reason, our pipeline first parses $(T,Y,\tilde{X})$, where $\tilde{X}$ is restricted to pre-treatment information and excludes treatment, outcome, and other post-treatment content. To quantify extraction reliability, we add the approximate rule-based check in **Table R3**. The results suggest that severe leakage is infrequent overall, and direct outcome leakage is particularly rare. This is not a formal measurement-error model, but it provides a practical check of extraction reliability. At the same time, we do not claim extraction is perfect: if relevant pre-treatment variables are missed, this is equivalent to incomplete observed confounding and may bias any extractor-style pipeline. We will clarify this limitation in the revision.
>
> **Table R3.**
>
> | Total N | Overall rule violation | Outcome leakage | Treatment leakage | Post-treatment leakage |
> | ------- | ---------------------- | --------------- | ----------------- | ---------------------- |
> | 35,528  | 535 (1.51%)            | 72 (0.20%)      | 397 (1.12%)       | 106 (0.30%)            |

---

> > ### Author Rebuttal · Reviewer_ca9u · 2026-04-03
> >
> > I am grateful to the reviewer for their reply, and the potential of the paper is acknowledged. In my humble opinion the extraction uncertainty and latent bias is a critical aspect that require further justification, relevant theoretical guarantees and discussion on how to diminish such bias / uncertainty will significantly increase the soundedness of the paper. Therefore I will remain the score.

---

> > > ### Author Response · Authors · 2026-04-06
> > >
> > > We appreciate your recognition of the paper’s potential, and we sincerely thank you for your careful follow-up and thoughtful engagement with both the paper and our rebuttal. We agree that **extraction uncertainty and bias are important practical risks in text-based causal inference**, and that the current manuscript should discuss them more explicitly as limitations.
> > >
> > > To make this limitation more explicit, we will add the following discussion in the revision:
> > >
> > > > Our method, like other extractor-style pipelines, depends on the quality of the upstream LLM extraction step. **First,** although SPIKED-CFR explicitly restricts $\tilde{X}$ to pre-treatment text and excludes outcome-revealing content in order to reduce leakage, residual treatment- or outcome-related signals may still remain in the extracted representation [1]. **Second,** relevant pre-treatment confounders may be missed or only weakly captured by the extraction process, so the learned representation should be viewed as an imperfect proxy rather than a complete adjustment set [2,3]. **Third,** extraction and representation learning may distort baseline information, which can create additional robustness and validity concerns in downstream causal estimation [4]. Recent work also shows that treating LLM-generated variables as error-free can bias downstream analysis and lead to overconfident conclusions [5]. **Improving the robustness of this upstream extraction stage therefore remains an important direction for future work.**
> > >
> > > Although **the primary scope of this paper is not to solve extraction error or hidden confounding left unresolved by LLM-based extraction**, we agree that this is an important practical issue and should be discussed more explicitly in the paper. **We will incorporate the above discussion into the revised “Limitations and Discussion” section** and make these limitations and their relation to the paper’s scope clearer. We again sincerely thank you for raising this point, as it will undoubtedly help us improve the paper’s clarity, completeness, and overall presentation!
> > >
> > > **References**
> > >
> > > [1] *Conceptualizing Treatment Leakage in Text-based Causal Inference*, NAACL 2022.
> > >
> > > [2] *Proximal Causal Inference With Text Data*, NeurIPS 2024.
> > >
> > > [3] *Challenges of Using Text Classifiers for Causal Inference*, EMNLP 2018.
> > >
> > > [4] *How to Make Causal Inferences Using Texts*, *Science Advances* 2022.
> > >
> > > [5] *Using Imperfect Surrogates for Downstream Inference: Design-based Supervised Learning for Social Science Applications of Large Language Models*, NeurIPS 2023.

---

### Decision · Program_Chairs · 2026-04-30

**Decision:**

Accept (regular)

**Comment:**

The proposed approach is aknowledged by reviewers as novel and technically sound. They appreciated the thourough experiments supporting the benefits of the chosen approach. Concerns where raised about the unverifiability of the assumptions on which the theoretical result rely on, and more generally the risk that various properties of the LLMs' latent representation may hinder the reliability of the approach. After considering the arguments exchanged during the discussion I recommend acceptance of the paper as the novelty of the proposed apporach and convincing experimental result on a very challenging problem bring a clear contribution to the community. I recommend to the authors to fully integrate the discussed limitations in the main text, such that the community can further address them in future work.